

# Glacier damage evolution over ice flow timescales

Meghana Ranganathan[1,2], Alexander A. Robel[2], Alexander Huth[3], and Ravindra Duddu[4]

[1]University Corporation for Atmospheric Research, Boulder, CO, USA
[2]School of Earth and Atmospheric Sciences, Georgia Institute of Technology, Atlanta, GA, USA
[3]NOAA Geophysical Fluid Dynamics Laboratory, Princeton, NJ, USA
[4]Department of Civil and Environmental Engineering, Department of Earth and Environmental Sciences, Vanderbilt University, Nashville, TN, USA

**Correspondence:** Meghana Ranganathan (miranganathan@uchicago.edu)

**Abstract.** The rate of mass loss from the Antarctic and Greenland ice sheets is controlled in large part by the processes of ice flow and ice fracture. Studies have shown these processes to be coupled: the development of fractured zones weakens the structural integrity of the ice, reducing ice viscosity and enabling more rapid flow. This coupling may have significant implications for the stability of ice shelves and the rate of flow from grounded ice. However, there are challenges with modeling

5   this process, in large part due to the discrepancy in timescales of fracture and flow processes as well as uncertainty in the construction of the damage evolution model. This leads to uncertainty in how fracture processes can affect ice viscosity and, therefore, projections of future ice mass loss. Here, we develop a damage evolution model that represents fracture initiation and propagation over ice flow timescales, with the goal of representing solely the effect of damage on flow behavior. We then apply this model to quantify the effect of damage on projections of glacier response to climate forcing. We use the MISMIP+

10  benchmark glacier configuration with Experiment Ice1r, which represents grounding line retreat due to basal melt forcing. In this model configuration, we find that damage can enhance mass loss from grounded and floating ice by $\sim 13 - 29\%$ in 100 years. The enhancement of mass loss due to damage is approximately of the same order as increasing the basal melt rate by $50\%$. We further show the dependence of these results on uncertain model parameters. These results emphasize the importance of further studying the multiscale processes of damage initiation and growth from an experimental and observational standpoint and of incorporating this coupling into large-scale ice sheet models.

## 1   Introduction

Viscous ice flow, which transports ice from ice sheet divides towards the ocean, is a significant control on the rate of ice sheet mass loss. The rate of ice flow through fast-flowing glaciers in Antarctica and Greenland has been increasing in the last two decades (Pritchard et al., 2009; Cook and Vaughan, 2010; Paolo et al., 2015; King et al., 2020; Wallis et al., 2023), enhancing

20  the contribution of ice sheets to global sea-level rise (Rignot et al., 2011; Fox-Kemper et al., 2021). Therefore, understanding and modeling the processes that govern the rates of ice flow is of particular importance.

The observed acceleration in ice flow has, in many cases, been linked to fracturing upstream from glacier termini. Lhermitte et al. (2020) and Sun and Gudmundsson (2023) have correlated recent accelerations of ice flow at Pine Island Glacier with the



development of fractures along the margins of the glacier. Surawy-Stepney et al. (2023a) similarly relates the development of
fracturing in Thwaites Glacier to changes in ice flow speed. These observations have been physically explained by fractures
structurally weakening the ice, thus reducing the effective bulk ice viscosity and enabling more rapid flow (Lhermitte et al.,
2020; Sun and Gudmundsson, 2023). Further, the accumulation of damage on ice shelves can reduce their ability to buttress
grounded ice, enabling more rapid flow from grounded regions to floating ice shelves (Borstad et al., 2013, 2016; Fürst et al.,
2016). Ultimately, the observed accumulation of fractures on both grounded ice and floating ice has the potential to affect ice
sheet mass loss through enhanced acceleration of flow and instability (Bassis et al., 2024), pointing to the need for fractures to
be included in ice sheet models used to predict ice flow changes.

Previous studies have taken different approaches to the modeling of fractures in ice sheets. Many studies have applied
fracture mechanics principles, including zero stress and linear elastic fracture mechanics (LEFM) theories, to understand the
propagation of crevasses leading to calving (Jezek, 1984; Van Der Veen, 1998a, b; Rist et al., 2002; Benn et al., 2007) and
to predict when large-scale rifts may propagate (Lai et al., 2020). However, LEFM is limited in its ability to represent the
effect of fracturing on viscous properties of a material. Firstly, defects within ice sheets range from microcracks and voids
($\sim 10^{-3} - 10^{-2}$ meters) to macroscale rifts and crevasses ($\sim 10^2 - 10^4$ meters) (Krajcinovic, 1996), and the grid size of
numerical ice sheet models is too large to explicitly capture the micro- and meso-scale fractures (Van Der Veen, 1998b; Jezek,
1984; Borstad et al., 2013). Secondly, fractures evolve on far shorter timescales ($\sim 10^0 - 10^4$ seconds) than ice sheet models
can resolve, as they are designed to model longer timescale ($\sim 10^0 - 10^5$ years) viscous deformation and so typically have time
steps of days to months. This discrepancy in timescales is a challenge this paper aims to tackle.

Other studies modeling ice fracture have applied the principles of continuum damage mechanics (CDM). Rather than mod-
eling the initiation and propagation of individual cracks, continuum damage mechanics uses a internal state variable to describe
the average density of defects within a representative volume of material, typically referred to as "damage" (Lemaitre, 1996;
Krajcinovic, 1996; Kachanov, 1999). This enables straightforward integration of fractures into constitutive models and pro-
vides a computationally efficient way of accounting for the effects of microdefects, thus allowing for a representation of the
effect of material damage on mechanical deformation across spatial scales (Krajcinovic and Mastilovic, 1995).

Most of the CDM modeling studies in the glaciology literature are focused on representing calving in viscous flow models at
the scale of an individual glacier or ice-shelf. Pralong and Funk (2005) originally proposed using a CDM model based on the
anisotropic theory for creep fracture developed by Murakami (1983), building on the works of Kachanov (1958) and Rabotnov
(1968), to simulate the calving process in an alpine glacier. This creep CDM model was later extended to include temperature
dependence (Duddu and Waisman, 2012), was implemented using nonlocal formulations (Duddu and Waisman, 2013a; Duddu
et al., 2013; Jimenez et al., 2017; Huth et al., 2021), and was combined with water pressure terms to represent hydrofracturing
(Mobasher et al., 2016; Duddu et al., 2020). Huth et al. (2023) applied the creep CDM model to the calving of Iceberg A68,
which calved in 2017, to demonstrate that it can replicate the rift propagation process during the two years prior to calving. As
another approach to representing damage in ice sheets, Borstad et al. (2012) applied an inverse method to estimate damage on
the Larsen B ice shelf prior to its collapse, as a way to constrain a calving law. A similar inverse technique was used to identify
the effects that damage has on weakening ice shelves (Borstad et al., 2013, 2016; Sun and Gudmundsson, 2023). Krug et al.




(2014) proposed a prognostic approach that uses a combination of CDM and LEFM models, which represents the weakening

of ice due to accumulated fractures and downward propagation of crevasses to eventually form rifts.

Alternatively to the creep CDM model, Bassis and Ma (2015) developed a ductile failure model to account for plastic necking and melting/re-freezing process, which can enhance basal crevassing. Sun et al. (2017) proposed a damage model that calculated the penetration depth of surface and basal crevasses, and coupled it to an ice sheet model to represent the evolution of crevasses within a shallow shelf approximation. Kachuck et al. (2022) described damage evolution using a combination of

the crevasse penetration depth model (Sun et al., 2017) and the plastic necking model (Bassis and Ma, 2015). Considering the relatively fast brittle fracture process in ice, Sun et al. (2021) and Clayton et al. (2022) developed phase field fracture models that intergrates LEFM and poromechanics concepts within the CDM framework, to describe hydrofracturing of crevasses.

While the approaches described above have been primarily applied to improve the understanding of calving processes, there are comparatively few studies that incorporate such damage models for the purpose of understanding the role of damage in

governing long-timescale (decadal to century-scale) changes in ice rheology and ice flow. Previously, Albrecht and Levermann (2012, 2014) presented a model framework for coupling damage evolution and flow and compared its results with present-day observations of Antarctic ice shelves. Lhermitte et al. (2020) and Sun et al. (2017) quantified the enhancement of flow due to damage evolution in forward simulations of an idealized glacier using the damage model developed in Sun et al. (2017). Despite this previous work, there remains significant uncertainty in the effects of damage on future flow, derived from the

choice of damage model, the choice of parameters in the damage model, and the applied forcing.

Here, we seek to quantify the effect of damage evolution on projections of ice loss. The specific goal of this work is to constrain the effects of damage-induced weakening on flow acceleration and ice loss. As such, this work does not aim to represent the effects of damage and damage-induced weakening on the localization and propagation of rifts leading to calving. We first propose a simplified model for incorporating scalar damage evolution into flow models that takes into account the

discrepancy in timescales between ice flow and ice fracture and reduces the parametric freedom within current damage models. We then apply this simplified model to a benchmark marine-terminating glacier configuration to quantify the effect of including damage into glacier flow with regard to ice mass loss.

## 2 Timescales of damage evolution

In the CDM framework, the density of fractures in a representative volume of material, generally called "damage" denoted by

$D$, can be simply treated as a scalar continuum state variable (Kachanov, 1958) and is constrained by $D \in [0, 1]$. If $D = 0$, the material is perfectly intact; whereas if $D = 1$, the material has lost all of its load-bearing ability due to the accumulation of fractures and other defects (Lemaitre, 1996).





The evolution of the damage variable in ice sheets is generally described through a transport (advection-reaction) equation in the Eulerian framework:

$$\frac{\partial D}{\partial t} + \mathbf{u} \cdot \nabla D = \begin{cases} f, & \sigma^{\dagger} \geq \sigma_t \\ 0, & \sigma^{\dagger} < \sigma_t \end{cases} \tag{1}$$

where $\mathbf{u} = (u, v, w)$ is ice velocity, $f$ is the damage evolution function describing the rate of change of damage due to deterioration or healing, $\sigma^{\dagger}$ denotes a chosen invariant of the Cauchy stress tensor $\boldsymbol{\sigma}$ (e.g., trace or maximum principal value), and $\sigma_t$ describes the stress threshold parameter above which damage begins to accumulate. While in general, anisotropic damage must be represented as a second order tensor, in this paper we will consider isotropic damage evolution, so damage is represented by a scalar variable. The choice of stress invariant and the stress threshold, as well as the nature of the source term, is not well-constrained and will be discussed further below. This representation uses a damage threshold based on stress, though other studies also use strain-based (Duddu and Waisman, 2013b) or strain-energy-based (Beltaos, 2002) thresholds.

In this study, we hypothesize that, for the purpose of evolving damage on long (flow) timescales, the specific nature of the damage evolution function $f$, which describes the small spatial- and time-scale processes of fracture, does not need to be explicitly modeled, thus eliminating $f$ and the parameters therein as poorly constrained parametric degrees of freedom. To test this hypothesis, below, we first conduct a scaling analysis. Next, we evaluate the timescales of fracture in a glacier flowline model, and use this analysis to construct a simplified damage evolution model. Finally, we evaluate applicability of this simplified model in different climate forcing scenarios.

## 2.1 Scaling Analysis

To understand the characteristic timescales of damage evolution, we nondimensionalize a one-dimensional form of Equation 1, based on the following scalings:

$$t = [t]t^* \tag{2a}$$
$$u = [u]u^* \tag{2b}$$
$$h = [h]h^* \tag{2c}$$
$$D = D^* \tag{2d}$$
$$f = [f]f^* \tag{2e}$$

wherein $t$ is time, $u$ is velocity, $h$ is ice thickness, $D$ is damage, and $f$ is the damage evolution function in Equation 1. Brackets denote the scaling for each parameter, and the asterisk denotes the nondimensionalization of each parameter. Using these scalings, we can rewrite Equation 1 in 1D as

$$\frac{1}{[t]} \frac{\partial D^*}{\partial t^*} + \frac{[u]}{[x]} u^* \frac{\partial D^*}{\partial x^*} = \begin{cases} [f]f^*, & [\sigma]\sigma^{\dagger *} \geq [\sigma]\sigma_t^* \\ 0, & [\sigma]\sigma^{\dagger *} < [\sigma]\sigma_t^* \end{cases} \tag{3}$$



As this problem is simplified to the advection equation with no source in the case of $[\sigma]\sigma^{\dagger*} < [\sigma]\sigma_t^*$, for the remainder of this derivation we drop this case and focus on the case in which damage accumulates. To identify characteristic timescales of this problem, we define the timescale of the flow problem $[t]$ as the advective timescale $[t_a]$:

$$[t_a] = \frac{[x]}{[u]}. \tag{4}$$

120   Applying this scales to Equation 3, we arrive at:

$$\frac{\partial D^*}{\partial t^*} + u^* \frac{\partial D^*}{\partial x^*} = [t_a][f]f^* \tag{5}$$

The units of $[f]$ is inverse time. Therefore, we can identify a nondimensional number

$$\delta = [t_a][f] \tag{6}$$

This nondimensional number $\delta$ can physically be interpreted as a ratio of timescales: $[t_a]$ is the advective timescale, and 125   $[t_f] = \frac{1}{[f]}$ is a fracture timescale, describing the timescale at which fractures initiate, grow, and coalesce. From Equation 5, when the rate of damage production is much greater than the advection rate, $\delta$ is large ($\delta >> 1$), and the advective timescale is far longer than the fracture timescale. When the rate of damage production is much less than the rate of advection, $\delta$ is small ($\delta << 1$). Therefore, the magnitude of $\delta$ is a key parameter that dictates whether damage accumulates very rapidly compared to the flow model timestep, or whether damage accumulates on a similar timescale as the flow model. We hypothesize that for 130   most typical damage models and ice flow timescales, $\delta >> 1$ and therefore damage accumulates faster than the flow timescale.

## 2.2   Flowline Model Coupled to Damage Model

To demonstrate the rapid rate of fracture within ice flow models, we couple a continuum damage mechanics model to a marine-terminating glacier flowline model. In the flowline model, the glacier terminates at the grounding line. The model solves the (nondimensionalized) one-dimensional shallow-shelf approximation momentum and mass balance equations, as in Schoof 135   (2007), with an implicit numerical scheme for velocity, thickness, and grounding line position, following a Weertman sliding law (Weertman, 1957) and power-law rheology with $n = 3$ (Glen, 1955). The bed is a linear function of the length of the glacier and the bed slope is prograde. The scales and relevant parameters used in the nondimensionalized flowline model and damage model are found in Table 1, and the full model equations can be found in the Supplement Section 1. The numerical scheme, following Schoof (2007), is available through an open-source repository (link in reference Robel, 2021), and was previously 140   applied in Robel et al. (2018) and Christian et al. (2022).

    Damage influences flow by the hypothesis of strain-rate equivalence, which states that if a damaged material produces a strain-rate response under an applied stress, then the same material with no damage produces the same strain-rate under an effective stress, which can be defined as $\tilde{\boldsymbol{\sigma}} = \dfrac{\boldsymbol{\sigma}}{1-D}$ (Lemaitre, 1985; Pralong and Funk, 2005). This is equivalent to introducing a scaling applied to the flow law prefactor $\tilde{A} = A(1-\bar{D})^{-n}$, where $\bar{D}$ is depth-averaged damage and $n$ is the viscous stress exponent, typically set to $n = 3$ (Glen, 1955; Cuffey and Paterson, 2010). To prevent the rate factor from becoming infinite, 145   the maximum value of $\bar{D}$ is set to $D_{\max} = 0.99$ rather than $D_{\max} = 1$.





**Table 1.** Scaling for the damage evolution model and flowline model

| Parameter | Value | Description |
|:---:|:---:|:---:|
| $[h]$ | 1000 m | Thickness Scaling |
| $[a]$ | 0.1 m yr$^{-1}$ | Accumulation Scaling |
| $[u]$ | $\left(\frac{\rho_i g[h][a]}{C}\right)^{1/(m+1)}$ m yr$^{-1}$ | Velocity Scaling |
| $[x]$ | $\frac{[u][h]}{[a]}$ m | Length Scale |
| $n$ | 3 | Flow Law Exponent |
| $A$ | $4.227 \times 10^{-25}$ Pa$^{-n}$ s$^{-1}$ | Flow Law Prefactor |
| $C$ | $7 \times 10^6$ | Sliding Prefactor |

We couple this flowline model to a CDM model. Equation 1 describes the general form of most CDM models, with an arbitrary damage evolution function $f$. Ultimately, the nature of the source term depends strongly on the specific physics of fracture mechanics represented in the model, and therefore the source term varies widely amongst damage models applied to ice sheets. In the coupling of the flowline model, we use the damage model proposed by Pralong and Funk (2005) for simulating glacier calving. The source term in this model was initially proposed by Kachanov (1958) and Rabotnov (1968) to describe the time-dependent accumulation of damage as a kinetic process, which occurs during the tertiary creep stage of a polycrystalline metals at high homologous temperatures. Thus, it was not intended to model specific fracture processes in ice sheets, as in Bassis and Ma (2015) and Sun et al. (2017), but rather to describe the bulk accumulation of creep damage in a representative material volume (Murakami and Ohno, 1981). A simplified version of this creep damage model uses a power law source term as follows:

$$f = B(\tilde{\sigma}^\dagger - \sigma_t)^r (1-D)^{-k} \tag{7}$$

where $B$ is a damage rate factor, $r, k$ are exponents, and $\tilde{\sigma}^\dagger$ is the chosen invariant of the effective stress tensor. The nondimensional parameter $\delta$ can be written as:

$$\delta = \frac{[x]}{[u]} B[\tilde{\sigma}]^r \tag{8}$$

To calculate stress for this continuum damage mechanics model, we extend the mesh at each gridpoint into the $z-$direction to capture depth-varying damage. This has been determined to be necessary for accurate representation of damage, particularly due to the effect of overburden pressure closing cracks (Keller and Hutter, 2014). Our method for calculating depth-varying stresses notably differs from that of Keller and Hutter (2014) in that we do not account for basal water pressure, which may allow for basal crevasses. Thus, the damage we estimate is purely surface damage. We explore the potential effects of incorporating basal crevassing in the Discussion section. The stress criterion we use for damage accumulation is the maximum (tensile) principal stress criterion $\sigma_1 \geq \sigma_t$. The scaling to nondimensionalize stress is based on the flow law (Glen, 1955):

$$[\sigma] = 2\tilde{A}^{-\frac{1}{n}} \left(\frac{[u]}{[x]}\right)^{1/n} \tag{9}$$





Damage is solved explicitly based on the velocity and thickness fields. In this implementation, the velocity and thickness is
solved with the damage field from the previous timestep, and damage increment at the current timestep is then calculated from
velocity and thickness at the current timestep. Further details of the numerical scheme can be found in the Supplement Section
1.

To run this coupled model, we initialize the coupled flow/damage simulation from steady glacier length of $\sim 450$ km long
and zero damage, and we evolve flow and damage together. We let the stress threshold for damage accumulation be $\sigma_t = 0.02$
MPa, an arbitrary value for the purposes of evaluating the timescales of damage accumulation. Constraints on this value will
be discussed later in this paper. With these parameters and those specified in Table 1, dimensionless parameter $\delta$ is $7.83 \times 10^3$,
with an advective timescale of $\sim 10^4$ years and a fracture timescale of $\sim 1.3$ years. The fracture timescale is significantly
smaller than the advective timescale, as hypothesized. We would thus predict that fractures accumulate much more rapidly
than they are advected by ice flow.

We evolve damage and flow with time steps of 1 month (Figure 1). Damage is produced in all regions where the maximum
Cauchy principal stress $\sigma_1$ equals or exceeds the stress threshold $\sigma_t$. The maximum Cauchy principal stress is equal to the
longitudinal deviatoric stress minus the overburden pressure, which is a function of the vertical position in the glacier $z$. In
this simulation, damage accumulates, as expected, primarily at the surface in the downstream region of the glacier, due to
high deviatoric stresses and low overburden pressure in that region. Within one month, $D \approx 0.2$ accumulates near the surface
(Figure 1a). Damage continues to accumulate each month until it reaches $D = 1$ everywhere where the stresses are large
enough for damage to initiate. This occurs within six months (Figure 1f). This agrees with the theory that where $\delta >> 1$,
fractures accumulate much more rapidly than they are advected away by ice flow. The implication of this is that the time scale
from damage initiation to saturation ($D \approx 1$) is on the order of months, and explicitly simulating this short transient growth
time scale has little effect on ice flow, which integrates the effects of $D$ on the effective viscosity over time scales of decades
to millennia. This will be shown explicitly in the next two sections.

### 2.3 Diagnostic Damage Model

Given the speed at which damage reaches its maximum, we propose a *diagnostic damage model*, which is valid in the cases
where $\delta$ is large and thus the fracture timescale is much less than the advective timescale. This diagnostic model has three
steps. (1) The model identifies regions where the chosen invariant of the effective Cauchy stress tensor $\tilde{\sigma}^\dagger$ meets or exceeds the
stress threshold $\sigma_t$, and it sets those regions of the domain to $D = 1$, as in:

$$D_{\mathrm{acc}} = \begin{cases} 1, & \tilde{\sigma}^\dagger \geq \sigma_t \\ 0, & \tilde{\sigma}^\dagger < \sigma_t \end{cases} \tag{10}$$

From $D_{\mathrm{acc}}$ (Equation 10), we find the depth-averaged damage $\bar{D}_{\mathrm{acc}_i}$, where $i$ denotes the current model timestep, by enforcing
the condition that depth-averaged damage cannot exceed some maximum value $D_{\max}$ (Equation 11a). This step accounts for
rapid accumulation of damage in regions where the stress exceeds the threshold for damage initiation. (2) The model advects
the depth-averaged damage field from the previous timestep $\bar{D}_{i-1}$ using a transport equation with no source term (Equation



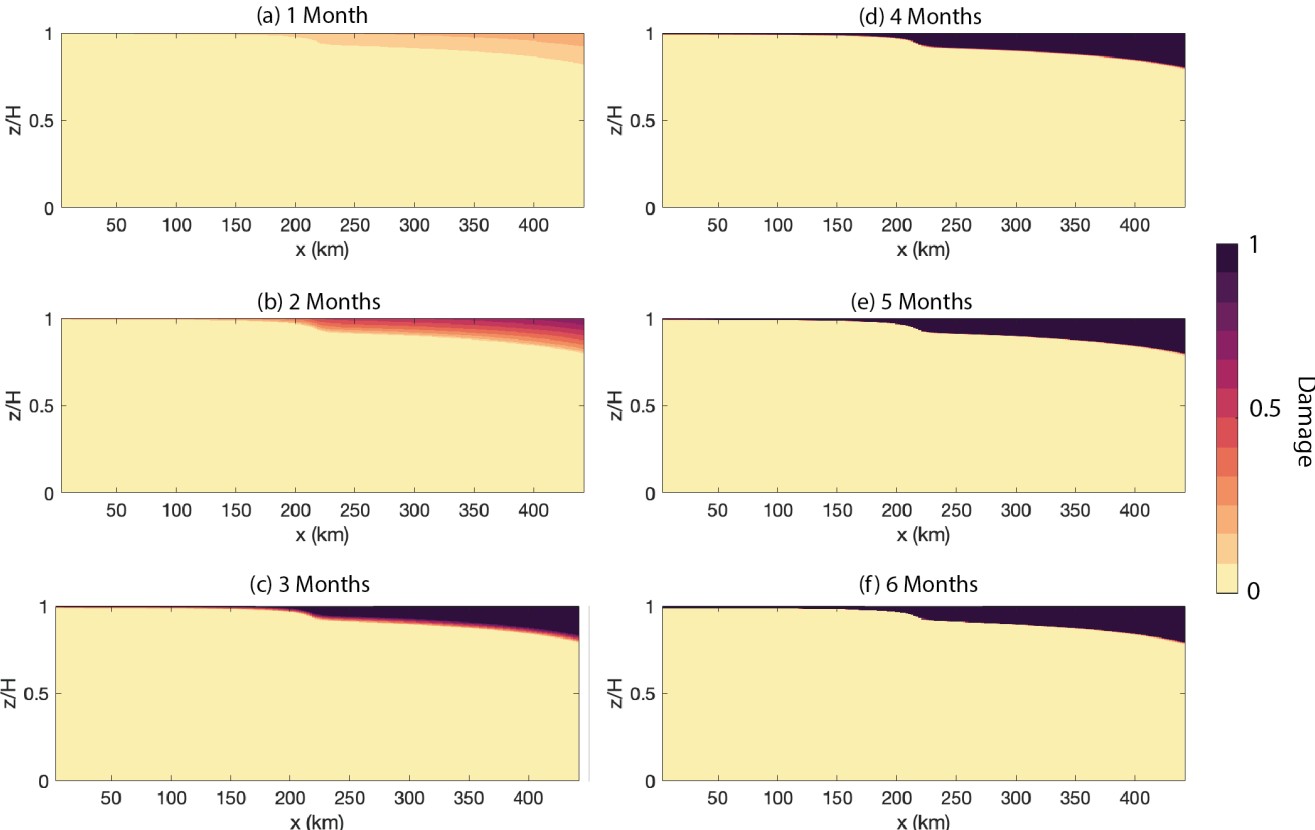

**Figure 1. Damage evolution in a flowline model over 6 months:** Damage fields from the coupled flowline/damage model with a timestep of 1 month for the first 6 months of the simulation. The flow moves from left to right, where the far right boundary is the grounding line and terminus position of the glacier and the far left boundary is the inflow boundary. The simulation initializes with zero damage and at a steady-state position. Damage accumulates to $D \to 1$ everywhere where stresses are high enough for damage to initiate within the 6 month period of the simulation.





11b) to produce a damage field $\bar{D}_{i-1}^+$, where $+$ denotes the solution of the advection equation at the current model timestep of the damage field from timestep $i-1$. This step allows for advection of damaged ice into regions that otherwise wouldn't initiate damage. (3) The model calculates the final damage field by taking the maximum of $\bar{D}_{\mathrm{acc}_i}$ and $\bar{D}_{i-1}^+$ (Equation 11c).

$$\bar{D}_{\mathrm{acc}_i} = \min[\bar{D}_{\mathrm{acc}_i}, D_{\max}] \tag{11a}$$


$$\frac{\partial \bar{D}_{i-1}^+}{\partial t} + \mathbf{u} \cdot \nabla \bar{D}_{i-1}^+ = 0 \tag{11b}$$

$$\bar{D}_i = \max[\bar{D}_{\mathrm{acc}_i}, \bar{D}_{i-1}^+] \tag{11c}$$

The goal of this diagnostic damage model (Equation 11c) is to represent the effect of damage on ice rheology and, thus,
projections of flow changes and mass loss through flow acceleration. This model cannot represent damage for the purpose of modeling calving or rift propagation in ice sheets, because such a goal would require small-timescale and small-spatial scale representation of fracture propagation and interactions with the local stress field. For the purposes of studying the effects of damage on ice viscosity and ice flow, this diagnostic damage model simplifies the problem of evolving damage in flow models by reducing the free parameters in the damage problem to a single uncertain parameter: the stress threshold, along with the
relevant form of the stress invariant dictating damage initation.

This diagnostic model has a similar basis to many existing, physically-based damage mechanics models. Sun et al. (2017) calculates the penetration depth of mode I basal and surface crevasses and advects these depths. A primary distinction between the model of Sun et al. (2017) and this diagnostic damage model is the stress threshold, as Sun et al. (2017) calculates crevasse depth using the assumption that crevasses penetrate to the depth at which the net effective stress is zero, thereby assuming
that the stress threshold is zero and allowing for basal crevassing. We treat the stress threshold as an unknown parameter and determine the sensitivity of our results to this parameter, and we do not account for basal crevassing. The diagnostic model produces a damage field that is similar in spatial extent to that of Sun et al. (2017), as shown in the Supplement Section 3. The diagnostic damage model is also expected to produce the same result of the continuum damage mechanics model applied in Pralong and Funk (2005), in the limit of the damage rate factor $B \to \infty$, when only surface crevassing is considered.

This model is valid for only specific advective and fracture timescales for which $\delta$ is large. Figure 2a shows the parameter space of $\delta$ for varying advective timescales and fracture timescales. In these limits, $\delta$ varies from $\sim 10$ to $\sim 10^6$, with delta increasing along a diagonal such that $\delta$ is large for large advective timescales and small fracture timescales. To show the range of values of $\delta$ for which this diagnostic damage model replicates full damage models, we compare the diagnostic damage model to the transient model of Pralong and Funk (2005) after 1 model year for the range of $\delta$ values in Figure 2a. We calculate
the percentage difference in grounding line position (scaled by the transient model position) after 1 year. The difference in grounding line position is $< 0.005\%$ for $\delta$ larger than approximately $10^4$ and increases to $> .03\%$ for $\delta > 10^2$ (Figure 2b). The absolute differences here remain small for all $\delta$ values due to the timescale of the model run (1 year). We show in the next section how these errors change for longer timescales and if we include climate forcing.





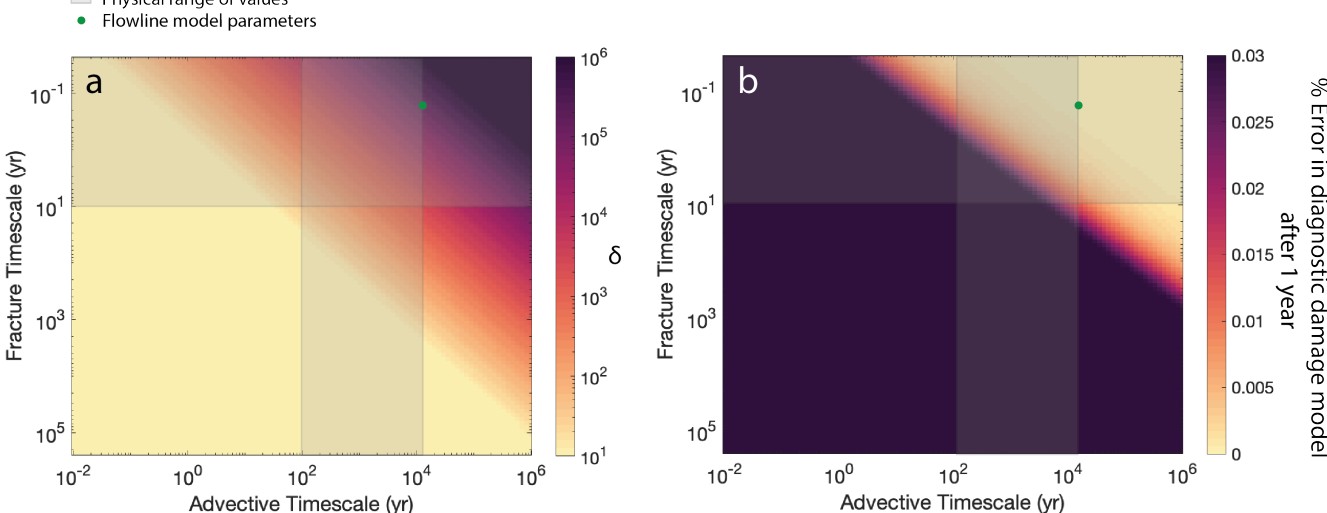

**Figure 2. Timescales for which the diagnostic damage model is applicable:** Parameter space of fracture timescale and advective timescale for (a) nondimensional parameter $\delta$ (Equation 8), (b) the error in the grounding line position using the diagnostic damage solver (the difference between grounding line position found from coupling the flowline model with the transient Pralong and Funk (2005) damage model and the grounding line position found from coupling the flowline model with the diagnostic damage model) after a simulation time of 1 year. The grey shading represents likely range of values of advective and fracture timescales based on geometry of ice streams and the physics of fracture, and the green dot denotes the parameters used in the flowline model.

## 2.4 Errors in Diagnostic Damage Model on Long Timescales

To demonstrate the accuracy of the diagnostic damage model (as compared to the full transient model) for longer glacier simulations, we compare the flow behavior using the diagnostic model and the full transient damage model. We initialize the flowline model with a steady state model configuration, and then we turn on damage coupling and evolve the model to a new steady state. We do so for different climate forcing simulations, as climate forcing can vary on timescales much smaller than the advective timescale.

We first test a case without climate forcing (Figure 3a), evolving the glacier state over $10,000$ years. Both damage simulations produce significant grounding line retreat of $\sim 6\%$ of the initial glacier length, with the control simulation ("No Damage") producing no change as the model is initialized at a steady-state. The diagnostic model replicates well the transient model behavior (Figure 3a,ii), with the difference in the grounding line position not exceeding $1\%$ of the deviation from the initial state (Figure 3a,iii).







**Figure 3. Comparison between transient and diagnostic damage models:** We run three different model simulations using no damage coupling, the full transient damage model of Pralong and Funk (2005), and the diagnostic damage model presented in Equation 11c and we present the grounding line position (ii) and the difference in grounding line position between the diagnostic damage model and the transient damage model (iii), including the difference in grounding line position as a percent of the amount of grounding line retreat by the transient model: (a) no climate forcing, in which the model is run for 10,000 years during which the idealized glacier retreats for the cases of damage coupled with the transient model and the diagnostic damage model. The diagnostic damage model produces very little deviation from the full transient model, (b) annual basal melt forcing, in which the model is run for 1,000 years. The idealized model retreats in all cases, though retreats faster and further in the case of damage coupling, (c) a warming climate in which basal melt forcing increases linearly over 1,000 years. The diagnostic damage model again reproduces the transient behavior well, with errors $< 5\%$ of the amount of grounding line retreat from the transient model.



Next, we test the diagnostic damage model in a simulation with annual variations in frontal melt forcing over 1000 years (Figure 3b). We define the following melt parameterization to simulate annual forcing:

$$\dot{m} = \mu_m + \eta_m \tag{12a}$$

$$\eta_m = M \sin\left(\frac{2\pi t}{N}\right) \tag{12b}$$

where $\mu_m$ is the mean melt rate, $\eta_m$ is the melt rate anomaly, $N$ is the period, $M$ is the amplitude of forcing anomaly, and $t$ is the time. We set $\mu_m = 15 \text{ m yr}^{-1}$, the period $N = 1$ year, the amplitude $M = 0.9\mu_m$ (Figure 3b,i). In all cases, incorporating damage produces more grounding line retreat than the control ("No Damage") simulation, suggesting that in this idealized setup, damage alters mass loss by acceleration of flow even when calving is not considered.

While the diagnostic damage model still produces roughly similar behavior to the full transient model (Figure 3b,ii), the difference between the two models is larger than in the case without melt forcing (Figure 3b,iii). However, the difference in grounding line position between the transient and diagnostic models remains under $5\%$ ($< 2000 \text{ m}$ of grounding line position difference). Further, most ice sheet simulations only run for a few centuries, in which case the differences in grounding line position remain between $2 - 3\%$, well within other sources of uncertainty in the grounding line position.

The final case we test is one with a long-term warming trend, in which the melt rate increases linearly over the course of the run (Figure 3c). In this case, we adjust Equation 12a to be:

$$\dot{m} = \mu_m + 0.2t \tag{13}$$

with $t$ in years (Figure 3c,i). The difference in grounding line position between the diagnostic damage model and transient damage model is comparable to the case of annual melt forcing, with the error remaining below $4\%$ for the run of 1000 years (Figure 3c,iii). In the supplement (Figure S1), we show other tests against interannual forcing, varying strength of the melt rate anomaly, and varying mean melt rate. Amongst all the runs, the diagnostic damage model well replicates the transient behavior with grounding line position differences of just a few percent of the overall change from the initial steady state.

This model does not include explicit healing processes in the form of damage sinks in the damage evolution equation. The only process preventing damage represented here is the effect of overburden pressure, which prevents cracks from propagating all the way through the ice thickness. The mechanisms of other healing processes are relatively uncertain, and many models do not include healing parameterizations due to uncertainty in the underlying physics and the form of the parameterization as applied to ice (e.g., Sun et al., 2017; Duddu et al., 2020; Huth et al., 2021). The models that do include healing do so by defining an arbitrary healing rate and applying this to stress or strain-rate (Pralong and Funk, 2005; Albrecht and Levermann, 2012, 2014). Other models can result in a reduction in damage due to physical processes (Bassis and Ma, 2015). Though we do not include similar parameterizations here, we could accomplish a simple parameterization by reducing the damage at each timestep by some fraction. Barring further physical intuition of experimental or observational data to inform the magnitude of healing, we leave an exploration of healing for future work. The estimates presented in this study, therefore, could be thought of as the upper bound on the effect of damage on flow.



In the next section, we apply this diagnostic damage model to a benchmark glacier simulation in a 2D ice sheet model, forced by basal melting, and quantify the effects of damage and damage evolution on the projections of future glacier behavior. We then explore the sensitivity of this coupled damage/flow model to the type of the stress criterion and threshold.

## 3  Effect of damage on flow

To identify the effect of damage on glacier flow, we set up an idealized 2D glacier, following the geometry of the benchmark glacier experiment MISMIP+ (Figure 4) (Asay-Davis et al., 2016). We use the open-source, Python-based ice sheet model icepack, which solves the shallow stream approximation of the momentum balance equations for glacier velocity (Shapero et al., 2021). The glacier is $640 \text{ km}$ long and $80 \text{ km}$ wide. The bed has a trough down the centerline, with the bed elevation higher in the margins (Figure 4a). We apply a Weertman-style sliding law and a power-law rheology, with parameters prescribed by Asay-Davis et al. (2016), and we initially evolve the glacier to a steady state over $10,000$ years, with a refined mesh in the vicinity of the grounding line as done in Shapero et al. (2021). The steady-state ice thickness varies from $\sim 1500 \text{ meters}$ upstream to $\sim 300 \text{ meters}$ on the ice shelf (Figure 4c). The glacier goes afloat (i.e., has a grounding line) approximately $460$ km downstream in the centerline, and the glacier velocity increases sharply downstream of the grounding line, from $< 200 \text{ m}$ $\text{yr}^{-1}$ to $> 700 \text{ m yr}^{-1}$ (Figure 4d).

We conduct the MISMIP+ experiment Ice1r, which simulates transient melt-driven grounding line retreat over 100 years. We initialize the model with the steady-state thickness and velocity fields and initiate melt according to a depth-dependent melt parameterization prescribed by Asay-Davis et al. (2016) (Equation 14; Figure 4b).

$$\dot{m} = \Omega \tanh \frac{H_c}{H_0} \max(z_0 - z_d, 0) \tag{14}$$

$\Omega$ is a free parameter that controls the magnitude of melting ($\Omega = 0.2 \text{ yr}^{-1}$ as in Asay-Davis et al. (2016)). $H_c = z_d - z_b$ is the difference between the ice draft (thickness of the ice below the water level) $z_d$ and the elevation of the bed $z_b$. We set $H_0 = 75 \text{ m}$ to be the reference thickness of the ice shelf cavity and $z_0 = -100 \text{ m}$ to be the depth at which basal melting starts. Most of the melt is concentrated near the grounding line, with a maximum melt rate of $\sim 75 \text{ m yr}^{-1}$ a few tens of kilometers downstream of the grounding line, beyond which the melt rate tapers off to $\sim 10 \text{ m yr}^{-1}$. This melt forcing is applied at each model timestep.

Damage is calculated using the diagnostic damage model (Equation 11c) presented in the previous section. Since this is a two-dimensional simulation, we apply the Hayhurst stress criterion to account for multiaxial stresses, in agreement with previous work (Murakami and Ohno, 1981; Pralong and Funk, 2005; Jimenez et al., 2017; Huth et al., 2021), defined as

$$\chi = \alpha \tilde{\sigma}_1 + \beta \sqrt{3 \tilde{\sigma}_{\text{eq}}} + (1 - \alpha - \beta) I_{\tilde{\boldsymbol{\sigma}}} \tag{15}$$

where $\alpha, \beta$ are constant parameters, and $\tilde{\sigma}_1$ is the maximum principal value, $\tilde{\sigma}_{\text{eq}} = \sqrt{\frac{1}{2} \tilde{\boldsymbol{\sigma}} : \tilde{\boldsymbol{\sigma}}}$ is the equivalent measure (the second invariant), and $I_{\tilde{\boldsymbol{\sigma}}}$ is the trace (the first invariant) of the effective Cauchy stress $\tilde{\boldsymbol{\sigma}}$. Note that the second term in the Hayhurst stress is the effective von Mises stress scaled by $\beta$.



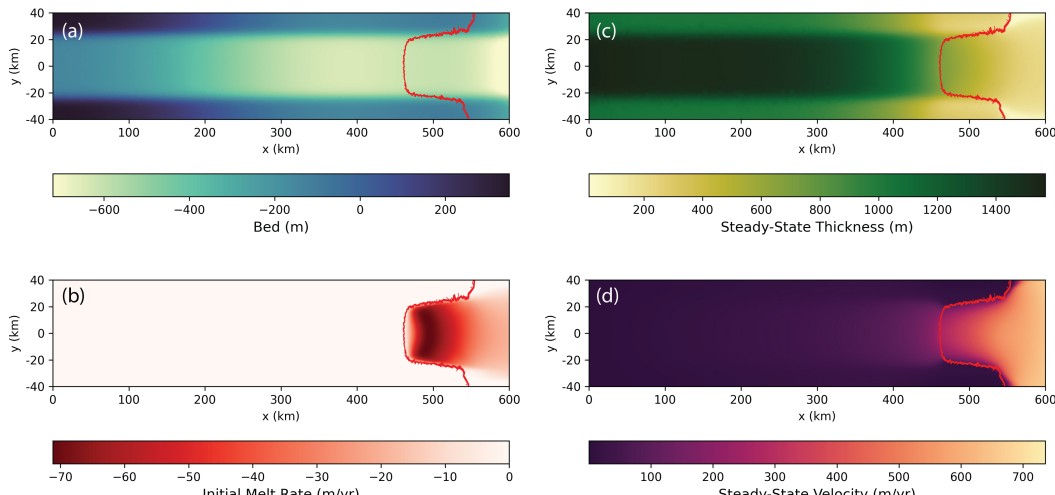

**Figure 4. MISMIP+ Glacier Simulation Setup:** Model geometry setup as prescribed by the MISMIP+ configuration (Asay-Davis et al., 2016), (a) Bed elevation, remaining constant in the model simulations, (b) Basal melt rate for $\Omega = 0.2 \text{ yr}^{-1}$, which is used initially. After $t = 0$, the melt rate is recalculated based on the updated geometry of the ice shelf, (c) ice thickness after a spinup of $10,000$ years to approximate steady-state, (d) ice velocity after a spinup of $10,000$ years to approximate steady-state. During spin-up, there is no damage accumulation or evolution.

While icepack solves the shallow stream equations in 2D, to calculate damage we follow the workflow of Keller and Hutter (2014) and Huth et al. (2021). We extrude the mesh, and subsequently project the 2D velocity field onto the 3D mesh, solely for the calculation of the Cauchy stresses, and we then calculate damage in 3D. We note that in 3D cracks are suppressed under high overburden pressure, so in most of the regions where $\bar{D} < D_{\max}$, the 3D damage field is non-zero only at the surface. Thus, we only capture surface cressvassing, but we explore in the Discussion section the potential effect of basal crevassing. We depth-average damage for the purposes of inserting damage back into the depth-averaged velocity solver. For the sake of the shallow flow solver, we only represent the effect of damage on deviatoric stresses (as in Jouvet et al. (2011); Krug et al. (2014); Keller and Hutter (2014); Sun et al. (2017); Huth et al. (2021)). To incorporate the effect of damage on pressure, we would need to apply a more sophisticated pressure boundary condition in the fully-damaged elements corresponding to open rifts. We anticipate that this effect on flow rate would be less significant compared to its effect on local stress field. In the figures for the remainder of this paper, we present depth-averaged damage $\bar{D}$, which is the field that is inserted into the flow solver in the next time-step.





**Table 2.** Model parameters used in the MISMIP+ simulations

| Parameter | Value | Description |
|:---:|:---:|:---:|
| $\alpha$ | 0.21 m | Hayhurst stress parameter (Pralong and Funk, 2005) |
| $\beta$ | 0.63 m | Hayhurst stress parameter (Pralong and Funk, 2005) |
| $\sigma_t$ | 0.1 MPa | The stress threshold; used as stated here unless otherwise denoted |
| $D_{\max}$ | 0.79 | The maximum value of damage; used as stated here unless otherwise denoted |
| $\Omega$ | 0.2 yr$^{-1}$ | Basal melting parameter (Asay-Davis et al., 2016); used as stated here unless otherwise denoted |
| $a$ | 0.3 m yr$^{-1}$ | Accumulation rate (Asay-Davis et al., 2016) |
| $C$ | 0.01 MPa m$^{-1/3}$ yr$^{1/3}$ | Friction coefficient (Asay-Davis et al., 2016) |
| $n$ | 3 | Stress exponent (Asay-Davis et al., 2016) |
| $A$ | 20 MPa$^{-n}$ yr$^{-1}$ | Flow rate parameter (Asay-Davis et al., 2016) |

To implement the Hayhurst stress criterion, we take the values of parameters from Pralong and Funk (2005), which are determined based on uniaxial experiments by Mahrenholtz and Wu (1992), and we present the relevant parameters in Table 2. Unless otherwise denoted, we use a stress threshold of $\sigma_t = 0.1$ MPa, in accordance with observational studies evaluating the stress threshold for fracturing in ice sheets (Vaughan, 1993; Grinsted et al., 2024).

The maximum value of 2D damage $D_{\max}$ ought to be set as close to 1 as possible, to simulate the complete loss of load-
bearing ability of the ice. Due to challenges with numerical convergence of the velocity solver, however, $D_{\max}$ must be set to be some value less than 1 to avoid the loss of ellipticity when solving for the velocity field ($D = 1$ implies effectively zero viscosity of ice). Here, we set $D_{\max} \approx 0.8$, which is the largest value for which we can consistently get numerical convergence. Because of the flow-rate parameter's nonlinear dependence on $D$ (viscosity $\propto (1-D)^{-3}$), the sensitivity of flow projections to $D_{\max}$ for values greater than $D_{\max} = 0.5$ in this MISMIP+ setup is quite small, as shown in the Supplement (Figure S6). This
may be because for $D_{\max} = 0.5$, ice viscosity in the maximally damaged regions is sufficiently close to 0 to produce similar flow behavior as those with $D_{\max} > 0.5$. However, it is presently unclear if the insensitivity to $D_{\max}$ is a robust behavior for all model geometries (e.g., Huth et al., 2023), so we recommend testing for $D_{\max}$ sensitivity in different model configurations. Since the diagnostic damage framework is only applicable when the fracture evolution rate is much smaller than the rate of advection, we validate these estimated damage fields in 2D against the full transient model of Pralong and Funk (2005) (Figure
S2) and previous studies considering damage in the MISMIP+ simulation (Figure S3). The results of this validation are shown in SUpplement Section 3.

In the remainder of this section, we seek to answer two questions: (1) what is the effect of initializing and evolving damage on the glacier flow response to forcing, compared to a control simulation where there is no damage in the model at all, and (2) what is the effect of evolving damage on this response to forcing, compared to a control simulation where damage is initialized
(such as by an inverse method) but not evolved? We present two metrics for the response of the glacier to forcing and damage: ice volume loss, which is the difference in total ice volume from the beginning of the simulation to a given timestep, and



grounded area loss, which is the difference in the area of grounded ice from the beginning of the simulation to a given timestep. In the subsequent sections, we present results from these two experiments.

### 3.1   Impact of damage production and evolution

We initialize the glacier simulation according to Figure 4, with zero initial damage, apply the melt rate beginning at $t = 0$ from Equation 14, and run the simulation for 100 model years in two simulations. For the "Without Damage" simulation, damage is zero everywhere and does not change, and ice viscosity remains constant (as listed in Table 2) for the 100 model years. For the "With Damage" simulation, beginning at time $t = 0$, damage is calculated by the diagnostic damage model (Equation 11c) at every model timestep, and this damage is used to calculate ice viscosity in accordance with the strain rate equivalence principle

$\tilde{A} = A(1 - D)^{-n}$. This is an input into the flow solver for the next timestep.

Damage initiates in two regions of concentrated damage in the margins just downstream of the grounding line, primarily due to elevated shear stresses concentrated in the margins and tensile stresses in center. Over 100 years, the lobes of damage on either margin extend towards the centerline, eventually connecting by $t = 100$ years (Figure 5a). The loss of load-bearing capacity in a continuous region across the ice shelf means that downstream ice transmits no buttressing stress upstream, thus

representing the dynamic effect of calving on the remaining ice, even in the absence of explicitly simulated iceberg detachment. Some damage also accumulates in the margins near the grounding line and in the center of the ice shelf. In these regions, $\bar{D} < 0.2$.

Including damage evolution in this model simulation causes both enhanced ice thinning and ice acceleration, concentrated around the regions of damage and the regions of maximum basal melting (Figure 5a,iv-v). Ice accelerates primarily near the

grounding line and towards the terminus of the glacier. There are also regions of acceleration in the trunk of the ice shelf. This aligns with the regions of maximum damage and also shows evidence of damaged regions downstream affecting flow upstream through decreased buttressing stress. Including damage causes the largest increase in acceleration of $> 400 \ \mathrm{m \ a^{-1}}$ at the grounding line, where the flow is likely responding to a combination of local damage and the reduction in viscosity of the ice downstream on the ice shelf. The grounding line in the center of the ice shelf is also the region where damage contributes

to thinning of the glacier, producing $\sim 400$ m more thinning at the grounding line than the simulation without damage. The effect on glacier thinning extends approximately 100 km upstream of the grounding line across the glacier, including in the margins. This suggests a damage feedback that is induced by basal melting, in which basal melt triggers thinning of the ice shelf, which accelerates ice flow, which then produces damage via increased stresses, which itself accelerates flow. Given that the melt parameterization used here is a function of ice draft, the thinning of the ice shelf further concentrates the basal melting

in the regions of thinning, which continue to produce more flow acceleration and damage production.

In response purely to melting without any damage (the control simulation), the grounding line retreats $\sim 75$ km (Figure 5a,i), a loss of $3,400 \ \mathrm{km^2}$ of grounded ice area (Figure 5a,ii) and $5,400 \ \mathrm{km^3}$ of total ice volume (Figure 5a,iii). In response to melting along with damage initiation and evolution ("With Damage"), the grounding line retreats an additional $\sim 15$ km, for a total retreat of approximately 90 km (Figure 5a,i). The glacier loses $4,400 \ \mathrm{km^2}$ of grounded ice area (Figure 5a,ii) and $7,000$





**Figure 5. Effect of damage on glacier response to basal melt forcing:** Results from a 100-year simulation of the MISMIP+ model configuration for two experiments: (a) evaluating the effect of damage initiation and evolution, (b) evaluating the effect of just damage evolution. We show (i) damage fields and grounding line change, (ii-iii) effect on grounded area and ice volume, (iv-v) effect on change in ice velocity and thickness.





km³ of total ice volume (Figure 5iii). In this simulation, including damage leads to a roughly 29% enhancement in ice volume loss from the no-damage simulation.

## 3.2  Impact of damage evolution

To isolate the effects of evolving damage, rather than the combined effects of initiating and evolving damage, we conduct a second experiment in which we spin up a new model steady-state that includes damage. At the start of the experiment, there is
an initial damaged field in both simulations (Figure 5b,i). In this case, the "Initialized Damage" simulation calculates viscosity from this initial damage field, thereby initializing damage but not evolving damage as the stress field evolves in response to basal melt forcing. This approximates the standard approach in ice sheet modeling where ice viscosity is inferred from an observed velocity field (which presumably includes the effects of damage) through inverse methods, but then viscosity is then kept constant during the model run. For comparison, the "Damage Evolution" simulation initializes and evolves damage
according to to the diagnostic damage model (Equation 11c).

In Figure 5b,i, the steady-state includes significant damage ($\bar{D} \approx D_{\max}$) in the margins of the glacier just downstream of the grounding line and elevated damage in the center of the ice shelf, increasing towards the terminus of the glacier. Damage accumulates generally in the margins just downstream of the grounding line, where shear stresses are high, and then damage advects downstream. There are patchy regions of elevated damage in the margins near the grounding line on the grounded
region of the glacier but otherwise there is no damage on the grounded ice. After 100 years of damage evolution in response to basal melt forcing, the extensively damaged regions widen in the margins of the ice shelf. There is more damage ($\bar{D} \leq 0.2$) on the center of the ice shelf near the grounding line, extending over the grounding line to the grounded regions of the glacier. This is likely due to thinning that occurs in the center of the ice shelf due to the high basal melt rates in those regions (Figure 5b,v), which reduces the effect of overburden pressure closing cracks. There are also lobes of low damage along the margins
of the grounded glacier upstream from the patchy fractured regions. This is also likely due to the thinning that occurs in the margins around $350 - 425$ km along the glacier (Figure 5b,v).

The simulation with damage evolution produces more acceleration in ice velocity at the grounding line (a difference of $\sim 500$ m yr$^{-1}$) and more thinning (a difference of $\sim 400$ m), aligning with the regions of maximum basal melting. There is minor development of damage at the surface ($\bar{D} \sim 0.1$) in this region. Further, the development of damage both in the center and in
the margins of the ice shelf reduces its buttressing capacity, which can enhance ice velocity in the simulation with damage evolution.

In response to melting, in the Initialized Damage simulation, the glacier loses $3,300$ km² of grounded ice area (Figure 5b,ii) and $6,000$ km³ of total ice volume (Figure 5b,iii). In response to melting and damage evolution ("Damage Evolution"), the glacier loses $3,800$ km² of grounded ice area (Figure 5b,ii) and $6,800$ km³ of total ice volume (Figure 5b,iii), resulting
in a $\sim 13\%$ enhancement of ice volume loss due solely to damage evolution. This suggests that initializing damage (either explicitly or implicitly by viscosity) at the start of a model run, for example through the use of inverse methods (e.g., Borstad et al., 2013), does capture some of the damage effects but does not sufficiently replicate the evolving effects of damage in response to climate forcing.




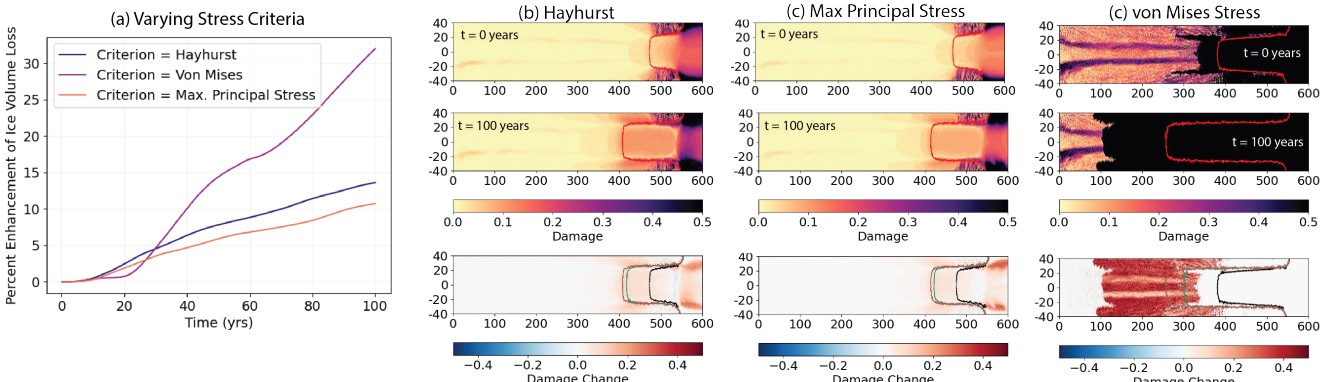

**Figure 6. Effect of damage on ice volume loss for varying stress criterion:** Results from a 100-year simulation for (b) Hayhurst criterion, (c) maximum principal stress criterion, (d) von Mises criterion. (a) We show the percent enhancement of ice volume loss (difference in ice volume loss between a simulation with damage evolution and a simulation with damage initialization but no evolution as a percentage of total ice volume loss in the damage initialization simulation) and the damage fields (initial and final damage, and total damage change).

### 3.3 Effect of uncertain parameters

A significant advantage of the diagnostic damage model is the simplicity of incorporating it into ice flow models. In particular, it reduces the number of free model parameters to one: the stress threshold, which encompasses both the specific value and the type of criterion used. However, there remains uncertainty in both the nature of the stress criterion and the value of the stress threshold. Here, we treat the type of stress criterion and the value of the stress threshold as uncertain model inputs and evaluate the effects of the choice of this stress threshold on projections of future flow (Figures 6–7). We set up model simulations to 415 study the effect of damage evolution, as in the previous section. To compare across different stress criteria (Figure 6) and thresholds (Figure 7), we report the percent enhancement of ice volume loss, which is the difference in the ice volume loss at each model year as a percent of the ice volume loss for the no-damage-evolution simulation at that model year. It can be described as the percent increase of ice volume loss that can be attributed to evolving damage during the simulation.

We first evaluate the effect of the stress criterion on future flow behavior (Figure 6). We explore three stress criteria applied 420 or studied in ice sheets previously: the von Mises stress, which describes the yielding of ductile materials (e.g., Albrecht and Levermann, 2012, 2014; Choi et al., 2018, 2021), the Hayhurst stress, as described previously (e.g., Pralong and Funk, 2005; Keller and Hutter, 2014; Mobasher et al., 2016; Jimenez et al., 2017; Huth et al., 2021, 2023), and the maximum principal stress. In conducting this comparison, we use a value of $D_{\max} = 0.5$ for numerical stability, as one of the criteria produces such extensive damage that the flow solver cannot converge with a larger value. We evaluate these three criteria using the same 425 stress threshold value of $\sigma_t = 0.1$ MPa. The von Mises stress criterion produces extensive full-thickness damage in its steady state across the ice shelf and near the grounding line on the grounded regions. There is also damage extending fully upstream in the glacier. As the glacier responds to basal melt forcing, the full-thickness damage continues to accumulate through the grounded ice regions. There is significant grounding line retreat in response to basal melting, with damage evolution producing





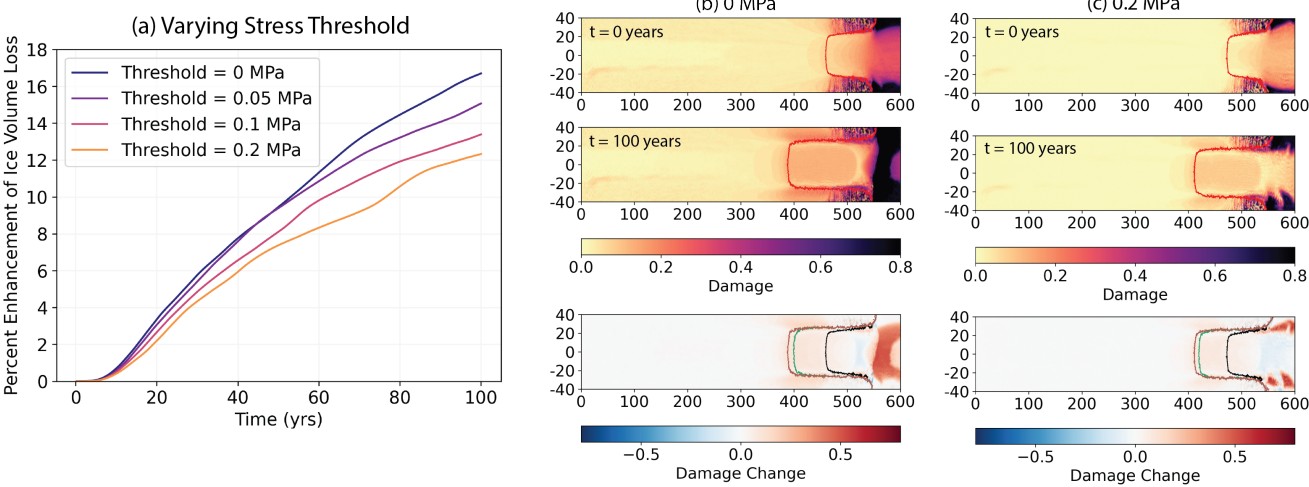

**Figure 7. Effect of damage on ice volume loss for varying stress threshold:** Results from a 100-year simulation for $\sigma_t = 0, 0.2$ MPa. (a) We show the percent enhancement of ice volume loss (difference in ice volume loss between a simulation with damage evolution and a simulation with damage initialization but no evolution as a percentage of total ice volume loss in the damage initialization simulation) and (b-c) the damage fields (initial and final damage, and total damage change) for $\sigma_t = 0, 0.2$ MPa.

far more grounding line retreat than without damage evolution. This is because the von Mises criterion is based on deviatoric
stresses, rather than Cauchy stresses, and thus it does not incorporate the effect of pressure preventing the propagation of cracks.
Using the von Mises stress criterion with damage evolution produces a $\sim 33\%$ enhancement in ice volume loss compared to
the "Initialized Damage" simulation. The maximum (most tensile) principal stress criterion produces the least enhancement
to ice volume loss ($\sim 11\%$) enhancement to ice volume loss compared to the simulation with no damage evolution. In this
case, the damaged regions are concentrated on the ice shelf, primarily in the margins. Finally, the Hayhurst criterion produces
a $\sim 14\%$ enhancement in ice volume loss by 100 model years. Given that the Hayhurst criterion is functionally a weighted
average of the von Mises criterion and the maximum principal stress criterion (along with a third term that is the first invariant
of the Cauchy stress), it is intuitive that this criterion produces behavior in between the other two criteria.

  We next evaluate the effect of the choice of stress threshold $\sigma_t$ (Figure 7). Varying the stress threshold produces less variation
in the ice volume loss enhancement. We treat $\sigma_t = 0$ MPa as an upper bound on the potential effect of damage and note that
this approximates a Nye zero stress model for fracture evolution in ice sheets. As the threshold increases, both the initial steady-
state damage field and the damage field after 100 years decreases, with the maximum damage concentrating in the margins.
As the threshold decreases, the damage field after 100 years of basal melting increases in the center of the ice shelf near the
terminus, and the full-thickness damaged regions in the margins thicken. Further, as the stress threshold value decreases, the
enhancement of ice volume loss due to damage evolution increases, from $10\%$ with $\sigma_t = 0.2$ MPa to $13\%$ with $\sigma_t = 0.01$
MPa. This suggests an uncertainty of only a few percent of ice volume loss enhancement due to uncertainty in the specific
value of the stress threshold, over the explored range. There is not a significant difference the ice volume enhancement between



$\sigma_t = 0.05$ MPa and $\sigma_t = 0.1$ MPa, due to there being few regions of stress separating these two thresholds in the idealized model setup. We expect that for a different model setup, there might be more of an enhancement for $\sigma_t = 0.05$ MPa than for $\sigma_t = 0.1$ MPa. In general, as the stress threshold decreases, there are more damaged regions and thus more of an effect on ice flow, but, importantly, there is an upper bound to this. The stress threshold cannot be smaller than 0, absent compressional fracture processes, and therefore the effect of damage on flow is bounded above, in this case at $\sim 14\%$ by 100 years.

## 4  Discussion

### 4.1  Magnitude of enhancement to flow due to damage

Based on the idealized simulations run with the benchmark MISMIP+ glacier geometry, we find that evolving damage enhances mass loss by $\sim 13\%$ compared to the simulation that initializes ice viscosity by damage but does not evolve damage. To contextualize the magnitude of ice loss enhancement, we compare this enhancement to that produced by increasing the climate forcing (in this case, the basal melt rate). We do so by varying the parameter $\Omega$ (as in Equation 14) and running simulations to 100 years in which we initialize a damage field but do not evolve damage during the simulation. We compare these to a simulation in which we initialize and evolve a damage field with a basal melting rate computed using $\Omega = 0.2$ yr$^{-1}$ (Figure 8). At year 100, the enhancement to ice loss and grounded area loss from damage evolution with $\Omega = 0.2$ yr$^{-1}$ is similar to increasing the rate of basal melt by $50\%$ in a simulation with no damage evolution. Basal melting causes a more immediate response in ice volume loss on timescales of $0 - 30$ years, while the effect of damage increases more significantly between $40 - 100$ years. Increased basal melting ($\Omega = 0.3$ yr$^{-1}$) and damage evolution with lower basal melting ($\Omega = 0.2$ yr$^{-1}$) have the same response to grounded area loss until approximately 60 years, at which point damage evolution produces slightly more grounded area loss ($\sim 100$ km$^2$ by year 100). Ultimately, these results highlight the significant effect that damage evolution may have on century-scale estimates of ice sheet change and provide motivation for the incorporation of a damage evolution model into large-scale ice sheet models.

### 4.2  Application of results to damage evolution in ice sheets

Many recent studies have presented observations of damage evolution in regions of the Antarctic Ice Sheet. Most notably, the southern margin of Pine Island Glacier in West Antarctica has accumulated significant damage over the last two decades (Lhermitte et al., 2020; Sun and Gudmundsson, 2023; Izeboud and Lhermitte, 2023). This damaged region initiates large-scale rifts that ultimately calve icebergs from the ice shelf (Lhermitte et al., 2020) and is correlated with a significant weakening (an increase in ice fluidity of approximately two orders of magnitude) in the shear margin and the outward movement of the shear margin (Sun and Gudmundsson, 2023). Similar damage evolution has occurred across the Thwaites Ice Shelf (Surawy-Stepney et al., 2023a; Izeboud and Lhermitte, 2023).

Our results here do not directly apply to the case of Pine Island Glacier, but we can provide insight into the processes occurring in the Amundsen Sea Embayment. In our idealized simulations, we identify similar weakening in the margins of the



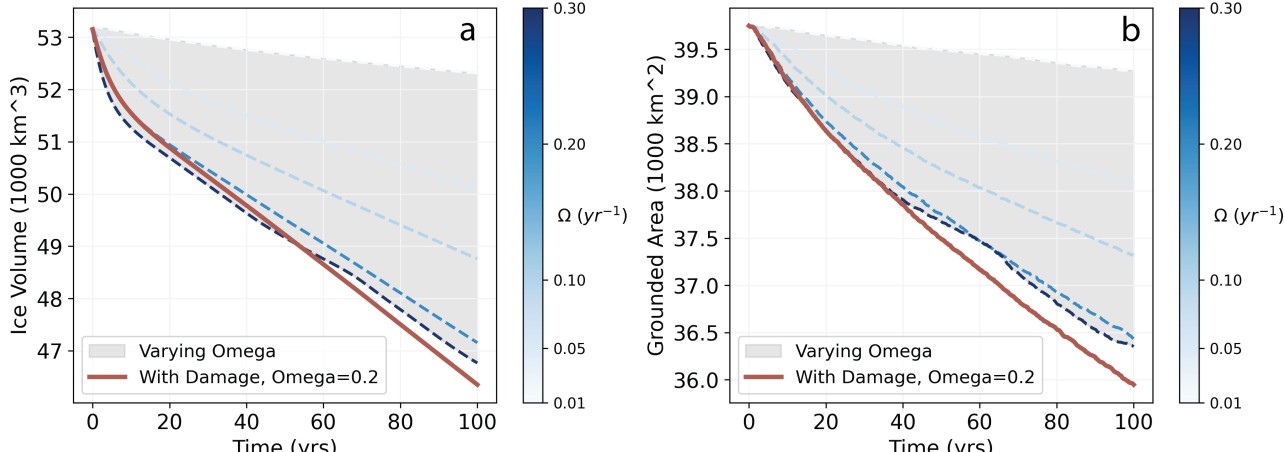

**Figure 8. Comparing the effect of damage on flow with the effect of increased basal melt rate:** Results of (a) ice volume, and (b) grounded area from 100-year simulations. The blue dashed lines show results from evolving the glacier in response to varying magnitudes of basal melting (in which the color of the line denotes to the magnitude of basal melt forcing) with no damage. The red line shows a case with damage evolution and $\Omega = 0.2\ \mathrm{yr}^{-1}$.

ice shelf and the initiation of maximally-damaged regions that extend across the width of the ice shelf, representing a calving event. These simulations do not represent margin migration, but future simulations can be set up to determine the extent of future margin migration due to damage evolution. Further, we identify a potential feedback between basal melting and damage evolution, in which basal melting causes thinning of an ice shelf, which enhances the spatial extent and the penetration depth of damage, which in turn enhances the thinning of the ice shelf. This is a feedback that may be taking place in these regions of the Amundsen Sea Embayment, such as Pine Island Glacier, where ocean warming is a primary driver of glacier change (Payne et al., 2004; Joughin et al., 2010). This may contribute to the processes driving damage evolution in the southern margin of Pine Island Glacier.

Finally, we identify the importance of constraining the stress criterion and stress threshold to reduce damaged-induced uncertainty in future flow estimates. Studies have made progress on constraining parameters in damage models. In particular, improved spatial and temporal resolution of satellite observations have enabled more high-resolution and reliable maps of fracture and damage fields across the Antarctic Ice Sheet (Vaughan, 1993; Hulbe et al., 2010; Lai et al., 2020; Izeboud and Lhermitte, 2023; Surawy-Stepney et al., 2023b; Grinsted et al., 2024). Since these fields are derived primarily from optical imagery of ice sheet surfaces, they are currently limited to damage extent on the surface. However, these observations can be used both to validate damage models and to provide insight into the values of $\sigma_t$ and the stress criterion used in damage models. Such observations have been applied to constrain the stress criterion for glacier ice against crevasses in the Antarctic and Greenland Ice Sheets, and they find that the von Mises criterion produces a generally good fit to observed crevasses, along with other criteria not considered in this study (Vaughan, 1993; Grinsted et al., 2024). Further, numerous laboratory and





observational studies have sought to quantify this value for ice, with laboratory estimates ranging from 800 kPa to 5 MPa (Currier and Schulson, 1982) and observational estimates ranging from 80 kPa to 1 MPa (Vaughan, 1993; Ultee et al., 2020; Grinsted et al., 2024).

## 4.3 Model assumptions and simplifications

We present and apply a novel diagnostic damage model in this study to evaluate the effect of damage on ice flow on long timescales. This model operates under the assumption that the timescales of ice flow are significantly longer than the timescales of ice fracture, and therefore damage accumulates rapidly compared to flow model timesteps. This model is not applicable, however, to simulations of ice dynamic processes occurring on short time scales and is therefore inappropriate to represent calving events or rift propagation on ice shelves, which can occur on timescales much less than a year (De Rydt et al., 2018;

Clerc et al., 2019; Cheng et al., 2021; Olinger et al., 2022; Surawy-Stepney et al., 2023a). Representing the coupled flow and fracture processes involved in rift initiation and propagation would require running transient damage mechanics and ice flow models on small timesteps that can represent the rift behavior on fracture timescales. To ensure numerical stability and mesh independence, previous work has shown that this benefits from nonlocal damage models (Jimenez et al., 2017; Huth et al., 2023) and a small enough timestep that damage accumulation does not exceed a set threshold value at each timestep.

Given these constraints to representing rift propagation, we suggest that the diagnostic damage model should mainly be used to model the effect of damage on long-term ice viscosity and flow evolution. There is still significant value in understanding and modeling the underlying mechanics of crack initiation and propagation on short time scales to answer other scientific questions; for example, the role of depth-varying ice material properties on crevasse propagation (Gao et al., 2023).

Even for modeling of long-term ice viscosity, there is a need for further research on the applicability of continuum damage

mechanics models to damage projections. Previous studies have noted that evolving damage over long timesteps can blur the sharpness of cracks or cause unrealistic crack propagation due to errors in the integration of the damage rate, which can result in unrealistically large regions of full-thickness damage (Huth et al., 2021). Transient damage models that aim to capture sharp rifting prevent this by adapting the timestep size so that changes in damage accumulation is sufficiently small over each timestep to ensure accurate and numerically-stable crack propagation (e.g., Huth et al., 2021, 2023). Evaluation of mesh- and

timestep-dependence (Supplement Figure S5) in this study has shown that, for the case of the MISMIP+ configuration, these issues likely do not affect the results presented here. However, for more complex glacier geometries and model configurations, there is a need for further examination of the timestep and mesh dependence of long-timescale damage modeling.

In this study, damage is estimated from 3D Cauchy stresses, which are found by taking the deviatoric stresses from the 2D flow model and subtracting the pressure from the overlying ice. Therefore, we are primarily modeling the effects of surface

crevassing, since the surface is where the ice overburden pressure is low enough to open cracks. However, there are crevasses that open up at the bottom of ice shelves, as the water pressure counteracts some of the ice overburden pressure (Weertman, 1969; Van Der Veen, 1998a; Luckman et al., 2012; McGrath et al., 2012; Buck and Lai, 2021). A way of modeling this effect of water pressure is to calculate Cauchy stresses from the deviatoric stress subtracted by an effective pressure, computed as the difference between ice pressure and water pressure for all depths below sea-level. This approach is outlined in Keller and Hutter





(2014) and Huth et al. (2021) and described further in Supplement Section 4. We show in the Supplement Section 5 that using effective pressure to estimate damage produces damage that extends further in ice thickness, including full-thickness damage in the margins near the grounding line. However, there remain significant uncertainties in the depth-variation of pressure. Most importantly, the estimation of stress here assumes an isothermal ice shelf, whereas in Antarctic ice shelves, the temperature of the ice likely increases with depth. This has implications for the stress field and thus the potential for basal crevasses to open,

as explored in Coffey et al. (2023). Therefore, we leave the exploration of basal damage and its effect on rheology and ice flow velocity for future work.

The diagnostic damage model assumes rapid damage accumulation but does not represent any processes that heal existing cracks or counteract the opening of cracks, with the exception of the effect of overburden pressure. A few ice damage models represent healing, which typically assume an arbitrary rate of healing (Pralong and Funk, 2005; Albrecht and Levermann,

2012, 2014) due to a lack of physical understanding surrounding healing processes in ice. Studies on other polycrystalline solids have represented kinetic healing processes, describing healing due to the movement of atoms closer together, by defining an activation energy for crack formation and healing (e.g., Miao and Engr, 1995; Arson, 2020)). Representing crack closure from overburden pressure by using three-dimensional Cauchy stresses is a different method of representing a similar mechanism, though it does not account for the effect of longitudinal, lateral, and shear deformation causing crack closure. Further work

needs to be done to understand the speed and magnitude of these healing processes in counteracting damage accumulation.

## 5 Conclusion

In this study, we seek to quantify the effect of damage on long-term glacier flow behavior. We first show that, in viscous glacier flow models, damage can be modeled diagnostically, where damage accumulation on short timescales is not explicitly modeled. We then apply this diagnostic damage model to quantify the effect of damage on marine-terminating glacier response

to climate forcing. We couple the diagnostic damage model to an ice flow model and force the idealized marine-terminating glacier with basal melting as in the MISMIP+ experiment Ice1r. We find that the reduction of ice viscosity due to damage that evolves during the simulation in response to the changing stress field enhances ice mass loss by $\sim 29\%$ compared to a simulation that does not consider the effect of damage on viscosity. We find that initializing ice viscosity by damage but not evolving damage in response to stress changes during the simulation captures some of this enhancement, but still evolving

damage with flow produces a $\sim 13\%$ enhancement in mass loss compared to solely initializing damage. This result suggests that initializing ice viscosity through inversions of damage (as in Borstad et al. (2012)) is necessary but not sufficient to capture the effects of damage on ice rheology.

The results of this work suggest that (1) the damage model presented here provides a simplified way of incorporating damage evolution into ice sheet models, as it does not require representation of specific fracture physics and reduces parametric uncer-

tainty within the damage model, and (2) incorporating the evolution of fractures across scales into ice viscosity is necessary to fully capture the effect of climate forcing on ice sheets on long timescales. However, there is still more to understand, from modeling, observational, and experimental perspectives, about the mechanisms of fracture accumulation and crack healing in





order to represent fully the effect of damage on ice flow in these models. There is also a demonstrated need for future obser-
vational and experimental constraints on the fracture threshold and fracture criterion, as well as more observations of fractures
across scales that can be used to benchmark damage models. The synthesis of such data with models that can represent the
effects of damage on long-timescale ice flow will be a significant step towards improving the physical fidelity of ice sheet
models.

*Code availability.* No new data was produced in this study. There were two models used, both of which are accessible via public repositories.
The flowline model, used in Section 2, is published in https://doi.org/10.5281/zenodo.5245271. The open-source ice sheet model icepack,
used in Sections 3 and 4, is available at the following: https://icepack.github.io/, along with the documentation and tutorials on use of the
model. The scripts for running both the flowline model and the icepack simulations, along with h5 files that contain the simulation output
for all of the icepack simulations presented in the main text and scripts to plot all of the figures in Sections 3 and 4, are available at
https://doi.org/10.5281/zenodo.11623054 (Ranganathan, 2024).

*Author contributions.* M.R. and A.R. conceived of the study and developed the methodology. M.R. conducted the simulations and carried
out the analysis. M.R. wrote the first draft of the manuscript, and all authors contributed to the analysis of results and the writing of the
manuscript.

*Competing interests.* The authors declare no competing interests.

*Acknowledgements.* The authors gratefully acknowledge Daniel Shapero and Andrew Hoffman for icepack support. This research was
supported by the NOAA Climate and Global Change Postdoctoral Fellowship Program, administered by UCAR's Cooperative Programs for
the Advancement of Earth System Science (CPAESS) under the NOAA Science Collaboration Program award #NA21OAR4310383. R.D.
acknowledges funding support from the NASA Cryosphere award no. 80NSSC21K1003 and the NSF Office of Polar Programs via CAREER
grant no. PLR-1847173.



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
