# Peer review of "Glacier damage evolution over ice flow timescales"

_EGUsphere, 2024_

## Referee Comment (RC1)

**Review of "Glacier damage evolution over ice flow timescales"**
**by Meghana Ranganathan et al.**

This study focuses on understanding the impact of glacier fracturing and damage on ice flow dynamics, particularly in the context of the Antarctic and Greenland ice sheets. The authors use a continuum damage mechanics (CDM) approach, representing damage as a scalar variable affected by stress and evolving over time (creation and advection of damage). They first propose a *"diagnostic damage model"* by assuming that the processes of damage creation occur on a short timescale with respect to the advection of this damage. They then validate their approach by showing that their hypothesis leads only to small error with respect to a *"transient damage model"* such as that of Pralong and Funck (2005). They then apply their model and compare it again with other implementations in the MISMIP+ glacier benchmark configuration, simulating grounding line retreat due to basal melt. Their results indicate that damage can enhance ice mass loss by 13-29% over a century (which would be equivalent to a 50% increase in basal melt rate), emphasizing the importance of including damage processes in large-scale ice sheet models for more accurate projections.

Overall, I find this paper to be well-constructed and clearly written, with figures that illustrate the main findings. The authors provide a detailed exploration of glacier damage evolution, and their model is applied thoughtfully within the MISMIP+ configuration to simulate grounding line retreat. While the quality of the work and the presentation are undeniable, I sometimes found it hard to switch from the main text to the supplement to get some specific information.

However, I have two primary concerns regarding the model implementation :

1. *Hypothesis for the diagnostic damage model, i.e., fractures accumulate much faster than they advect.*

   The choice of stress threshold in the model must have a significant impact on the results, as it directly influences where and how easily damage initiates. In the first experiment, the authors use $\sigma_t = 0.02$ MPa, a very low threshold that contrasts with higher values used in other studies (e.g., Krug et al., 2014; Sun et al., 2017), sometimes calibrated to match observed calving rates. While I understand that the model's focus is on damage accumulation rather than explicit calving, this low threshold promotes rapid damage development across broader ice regions than higher threshold values. This, combined with the feedback effect of damage on the source term (i.e., $f = f((1 - D)^{-k})$), means that initial damage quickly propagates in subsequent timesteps. Given these sensitivities, a discussion of the implications for using a low $\sigma_t$, as well as the role of the damage rate factor $B$, would help better justifying the hypothesis.

   Without this, as a reader, I feel like the value picked for $\sigma_t$ and $B$ (I could not find the value) affect the realism of the hypothesis $\delta \gg 1$, leading to fast damage creation rate. It makes it look like the experiment is built to match the hypothesis. While I do not see any argument to revoke the hypothesis, I do not think either that the results validate the hypothesis–although later experiments seem to show little effect of $\sigma_t$ which could be specific to the experiments? The sentence *"This agrees with the theory that where $\delta >$*

> 1,*fractures accumulate much more rapidly than they are advected away by ice flow*"
is therefore a bit of a stretch to me. Since this hypothesis and the first experiment
condition the definition of $D_{acc}$. I think that these limitations should be better discussed

2. *Unsymmetrical damage for a symmetrical geometry and numerical artifact:*

Since the MISMIP+ configuration is symmetrical along the central flowline, any
asymmetry in the modeled damage field (or other model output) may indicate numerical
artifacts rather than genuine physical behavior. Artifacts of this nature often arise from
issues like model stability  (e.g., CFL condition) or other challenges in the advection
scheme, particularly in handling diffusion-free processes like damage. In these cases,
artificial diffusion or other stabilization techniques are often used, which can introduce
numerical artifacts.

I would suggest the authors discuss whether such factors could contribute to any
asymmetry observed in their model results and clarify the type of stabilization methods
employed, as well as any expected artifacts. Addressing these potential sources of
numerical asymmetry (and diffusion) would strengthen numerical aspect of the study.

Regardless of these concerns, I believe this paper makes a valuable contribution to the field
and certainly merits publication after what I consider to be relatively minor revisions. The
study will be highly useful to the modeling community to integrate damage processes into ice
flow models.

**Specific comments**

- Line 17: I found this first sentence a little strange, I guess that the authors refer to
"viscous ice flow" as opposed to "elasto-brittle calving". Maybe consider rephrasing
this.
- Line 30: the term "instability" is not really clear here. I would detail a little more the
processes it refers to.
- Line 40: I would expand the ice sheet model timescale to as low as ~$10^{-1}$ year. More
and more models are interested in seasonal changes (without going to visco-elastic
models interested in tidal effect, …).
- Line 60:  I would change "accumulated fractures" for "accumulated damage" since
CDM was used to enhance damage, until damage was deep enough to trigger a
fracture over the entire column with LEFM.
- Line 72: switch the order of the two citations
- Line 102: "We evaluate the applicability"
- Line 106:  I think that TC writes equations as this: "Eq. (1)". Think about correcting this
for other references to your equations
- Equation 5: You therefore assume that $[t_a] = [t]$ to simplify your first term but isn't
$[t]$ the resultant of the advective ($[t_a]$) and the fracture ($[t_f]$) timescale? If guess you
assume $[t_f] \ll [t_a]$ to make the simplification but this relies on the hypothesis you do
after. I might be missing something here.
- Line 130: how do you justify this hypothesis for a typical damage model?

- Line 135: could you precise if you use a linear or a non-linear (typically $m = 3$) Weertman friction law. I could not find the information in the supplementary material.
- Line 174: you mention $\sigma_t = 0.02$ MPa as an arbitrary value. Other studies use much higher values (at least about 0.1 MPa) (e.g., Krug et al. 2014; Sun et al., 2017). For Krug et al. (2014), these values have been calibrated with observations of calving rate. Although I understand that your model only aims to simulate damage (without going to calving or rifting), I think that the choice of $\sigma_t$ is very important and will largely affects the result. Low $\sigma_t$ allows to easily damage the ice in many regions, and due to the term $(1 - D)^{-k}$ initiated damage leads to even more damage at the next timestep. It also depends on the value of $B$ in Eq. (8).
- Lines 216–224 and general statement about Section 3: I think that the limitation of your hypothesis is visible in your supplementary material where Damage seems particularly high and potentially overestimated. For example, your damage is much higher than Sun et al. (2017) but it might also be due to the difference in the criterion used to calculate the damage source () might also present biases due to other limitations in the physical model and its numerical implementation).
- You allow for your model to reach values of $D$ up to 0.99 to avoid null denominator in the effective stress model and numerical stability and convergence issues. However, is $D = 0.99$ a realistic value for CDM. While $D<1$ is a numerical condition to avoid infinite ice fluidity, I think that for too high damage values, especially when over large areas and deep into the column, the CDM really shows its limitations as we continue to simulate something that is far from being continuous as continuous. Later you mention in Section 3 that you set up $D_{max} \sim 0.8$. Can you precise if this is only for the MISMIP+ experiment and that you use $D_{max} = 0.99$ in the previous experiments?
- Line 230: I don't really understand the point of calculating the percentage difference in grounding line position after 1 year for different $\delta$. You then check the same error after a longer time (e.g., $10^2$– $10^4$ years) for which your diagnostic model is supposed to be more valid, which to me is a much better way to look at the "error" of the diagnostic model.
- Line 266: you mention healing as a form of damage sink in the advection equation. You might consider mentioning that the right end side of Eq. (1) is a "damage source/sink" when you present the equation.
- Line 286: Could you precise the resolution of the mesh in the vicinity of the grounding line?
- Line 310: I would add a comma: "We note that in 3D, cracks are …"
- Lines 310–315 (this could actually be a third main concern):

The model is Shallow Shelf (SSA) and therefore computes horizontal velocities considered constant over the ice column (plug flow). You can therefore assume null vertical velocities but this would greatly limit damage deep into the column (i.e., no vertical advection of the crevasses). You might also neglect "significant" vertical stress components, potentially leading to inaccurate stress calculations.

Assuming mass conservation, you can recompute vertical velocities (based on surface and basal accumulation/melt and some assumption on the distribution) but you would need to assume a vertical distribution. Could you give more detail here?

Since a lot of the damage creates close to the grounding line where the SSA solution and the resulting stress computation is more prone to errors, what impact do you think the Stokes approximation has on the damage solution (you mention the role of longitudinal, lateral and shear deformation on crack closure later but not on the creation of crevasses/damage them/itself)? This would be interesting to discuss.

- Line 336: "SUpplement" → "Supplement"
- Line 346: I suggest "… and run the model for 100 years in two simulations".
- instead of "… and run the simulation for 100 model years in two simulations" .
- Line 515: The blurriness of sharp cracks is also due to stabilization techniques for the advection equation. Stabilization techniques often rely on artificial diffusion, even for small timesteps and or small CFL numbers.
- Line 523: From this statement, I understand that there is no advection of damage in the column except for the fact that if the ice base is melted, a larger part of the column could be affected by damage? As I said in a previous comment, a part of the "vertical advection" of the damage could therefore be due to a numerical vertical diffusion of the solution. I think this could be better discussed here.

---

## Referee Comment (RC2)

**Review of Glacier damage evolution over ice flow timescales**

**General Comments**

This paper looks at how damage can be incorporated into ice sheet models. It seeks to quantify the importance of damage and its effect on mass loss from ice sheets. The damage is described by a scalar variable and incorporated into the ice sheet model in a continuum fashion, where the damage causes the ice viscosity to decrease. The authors suggest a simplified model for damage evolution, based on the assumption that the damage production timescale is much less than the timescale for advection. Hence, the ice can be assumed to be instantaneously damaged when the stress exceeds a certain threshold. This has the appealing advantage of removing many free parameters which occur in continuum damage evolution models, with little effect on the results as the mechanisms for damage production need not be modelled or parameterised. The results of this simplified model are first compared to the transient damage model to test this assumption. The model is then applied to the MISMIP+ benchmark to assess the impact of damage on grounding line retreat.

Generally, I think the paper is of very good quality and can be published subject to minor corrections.

**Originality**: The diagnostic model proposed is novel, and the new model is also applied to estimate the effect of damage on grounding line position and on ice mass loss over the next century.

**Scientific Quality**: The science is of good quality. The simplifying assumption is well tested with the non-dimensional analysis, though I would like to see more detail in the possible values of δ (see specific comment). Furthermore, the extension of the model to the MISMIP+ case serves as a good test, and they also link the impact of damage to final changes in mass loss. The discussion is well thought out and the conclusions are backed up by the main text.

**Significance**: The paper is significant for two reasons: firstly, proposing a novel diagnostic model for damage which should be easier to include in ice sheet models. Secondly, they also highlight the importance of damage on grounding line position and mass loss

**Presentation quality:** The paper is well written and concise. The figures are clearly explained in the text.

**Specific Comments**

As the other reviewer mentions, I think the authors should be clear on the uncertainties within their estimates of the advection and production timescales, and how does this propogate through to δ. What is the maximum and minimum value that δ can take given these uncertainties on the stress threshold, glacier length etc.? Where might this assumption not be valid?

The supplement is significant, and I think it would be helpful to move some of it into the main text. In particular, I think Fig S2 and the accompanying text would be helpful to further show that this diagnostic model is still valid in less idealised configurations.

Line 324: Am I correct in thinking that limiting the value of Dmax effectively limits the maximum softening caused by damage? Such that Dmax = 0.5 gives a max softening of 8 etc.? If so this may be helpful to explain in the text.

**Technical Comments**

Fig 5: The label for (b) (i) "Damage field after 100 years" is directly over a subplot for t=0 years: this label should be changed for part (b) of the figure.

Supplement Line 119: WhenIf

---

## Author Comment (AC1)

The comments from the reviewers are in black font and our responses are in blue font.

**1 Hypothesis for the diagnostic damage model, i.e., fractures accumulate much faster than they advect.**

The choice of stress threshold in the model must have a significant impact on the results, as it directly influences where and how easily damage initiates. In the first experiment, the authors use $\sigma_t$ = 0.02 MPa, a very low threshold that contrasts with higher values used in other studies (e.g., Krug et al., 2014; Sun et al., 2017), sometimes calibrated to match observed calving rates. While I understand that the model?s focus is on damage accumulation rather than explicit calving, this low threshold promotes rapid damage development across broader ice regions than higher threshold values. This, combined with the feedback effect of damage on the source term (i.e., $f = f((1 - D)^{-k})$), means that initial damage quickly propagates in subsequent timesteps. Given these sensitivities, a discussion of the implications for using a low $\sigma_t$, as well as the role of the damage rate factor $B$, would help better justifying the hypothesis.

Without this, as a reader, I feel like the value picked for $\sigma_t$ and $B$ (I could not find the value) affect the realism of the hypothesis $\delta \gg 1$, leading to fast damage creation rate. It makes it look like the experiment is built to match the hypothesis. While I do not see any argument to revoke the hypothesis, I do not think either that the results validate the hypothesis?although later experiments seem to show little effect of $\sigma_t$ which could be specific to the experiments? The sentence ?This agrees with the theory that where $\delta \gg 1$,fractures accumulate much more rapidly than they are advected away by ice flow? is therefore a bit of a stretch to me. Since this hypothesis and the first experiment condition the definition of $D_{acc}$ . I think that these limitations should be better discussed

We thank the reviewer for this point and acknowledge that some of this discussion was not clear enough in the initial submission. The thesis of the diagnostic model is that the rate of damage accumulation is rapid enough that it generally occurs within a model timestep and, therefore, does not need to be explicitly modeled. While we agree that the choice of the stress threshold significantly affects the extent of damage produced, the stress threshold does not play a role in the rate of damage production; this is modulated entirely by the two fracture parameters $B$ and $r$, as well as the stress scale (which depends, in the flowline model, on ice rheology, the velocity scale, and the length scale).

We also agree that the value of $\sigma_t$ is quite a bit lower than we would expect in natural ice sheets. This value was chosen because of the simplicity of the flowline model; it produces generally very low stresses, requiring a low stress threshold in order to produce fracture. Since we wanted to study the importance of the rate of damage accumulation, we had to pick a stress threshold that would produce some amount of damage. Since the value of the stress threshold does not appear in $\delta$, it does not affect the validity of the results. To show this, we will rerun Figure 2 for a different stress threshold value, $\sigma_t$ = 0.05 MPa (which we will include in the Supplement), which will show largely the same results as in Figure 2.

The parameter study in Figure 2 does, effectively, consider the role of $B$ (and all other parameters within the fracture timescale) on this hypothesis. If we use the Pralong and Funk 2005 fracture source term, the fracture timescale is defined as:

$$[t_f] = B[\tilde{\sigma}]^r \tag{1}$$

and therefore varying the fracture timescale effectively varies the fracture parameters, and the hypothesis that fracture accumulation is much faster than flow advection does not apply for low values of $B$. We present Fig. 2 in fracture timescale space, rather than $B$ space, to make it generally applicable to many

different damage models (that is, someone could calculate the fracture timescale of a different damage model and determine if this hypothesis is valid using Fig.2). Therefore, the applicability of the hypothesis is explicitly not dependent on any particular damage model but is instead presented as a more general theory.

The applicability, however, to a specific problem does require determining the fracture and flow timescales. Doing this, barring any intrinsic or physical understanding of fracture timescales, requires us to assume some damage model and calculate the timescale to verify where it falls in $\delta$ space. This is where we, for illustrative purposes, apply the Pralong and Funk model for comparison. The specific values of $B$ and $r$ are those widely applied in other glaciology studies building on the Pralong and Funk model [11, 7] and were originally constrained by laboratory experiments [9]. We will improve clarity of these points in the text in Section 2.

**2   Unsymmetrical damage for a symmetrical geometry and numerical artifact**

I would suggest the authors discuss whether such factors could contribute to any asymmetry observed in their model results and clarify the type of stabilization methods employed, as well as any expected artifacts. Addressing these potential sources of numerical asymmetry (and diffusion) would strengthen numerical aspect of the study.

Since the MISMIP+ configuration is symmetrical along the central flowline, any asymmetry in the modeled damage field (or other model output) may indicate numerical artifacts rather than genuine physical behavior. Artifacts of this nature often arise from issues like model stability (e.g., CFL condition) or other challenges in the advection scheme, particularly in handling diffusion-free processes like damage. In these cases, artificial diffusion or other stabilization techniques are often used, which can introduce numerical artifacts.

This is a great point, it is correct that these asymmetries do not arise from the model configuration, given that both the MISMIP+ geometry and the forcing are symmetric. While there are no stabilization techniques imposed in the solver, this asymmetry could be due to a number of factors. We agree that numerical diffusion does occur in Eulerian advection schemes, and small amounts of numerical diffusion could explain these asymmetries. This could also arise from physical symmetry breaking, in which asymmetric patterns can arise in systems that contain symmetric equations and forcing due to, for example, small errors from numerical approximations amplifying. Given that reductions in the timestep do not have significant effects on the solutions (Supplement Fig. S4), the solution appears stable towards numerics and thus this could be evidence for physical symmetry breaking explaining these asymmetries. This could also arise due to asymmetries in the mesh itself, which can be seen in some of the grounding line profiles. While we don't believe this affects the scientific conclusions of the paper, we agree that it is important to explain these artifacts. We will add a discussion of these asymmetries in Section 3 when discussing the MISMIP+ results. In the revision, we can also discuss methods for addressing these numerical issues, such as the Material Point Method employed by previous papers [6, 7, 8].

**3   Specific Comments**

Line 17: I found this first sentence a little strange, I guess that the authors refer to ?viscous ice flow? as opposed to ?elasto-brittle calving?. Maybe consider rephrasing this.

This sentence is intended to describe the viscous response of ice sheets, rather than brittle calving. We have simplified this to just say "Ice flow".

Line 30: the term ?instability? is not really clear here. I would detail a little more the processes it refers to.

We've specified more clearly what we mean in that line.

Line 40: I would expand the ice sheet model timescale to as low as ~ $10^{-1}$ year. More and more models are interested in seasonal changes (without going to visco-elastic models interested in tidal effect, ...).

Done.

Line 60: I would change ?accumulated fractures? for ?accumulated damage? since CDM was used to enhance damage, until damage was deep enough to trigger a fracture over the entire column with LEFM.

Done.

Line 72: switch the order of the two citations

Done.

Line 102: ?We evaluate the applicability?

Done.

Line 106: I think that TC writes equations as this: ?Eq. (1)?. Think about correcting this for other references to your equations

Done.

Equation 5: You therefore assume that $[t_a] = [t]$ to simplify your first term but isn?t $[t]$ the resultant of the advective ($[t_a]$) and the fracture ($[t_f]$) timescale? If guess you assume $[t_f] \ll [t_a]$ to make the simplification but this relies on the hypothesis you do after. I might be missing something here.

The timescale $[t]$ that we had defined in the first submission was describing the timescale of the flow problem, rather than the fracture problem, and therefore was by definition the advective timescale. In this revision, we have defined the timescale of the flow problem up front and removed any use of $[t]$, to avoid confusion.

Line 130: how do you justify this hypothesis for a typical damage model?

Thanks for this point; we acknowledge that this should have been made far more explicit in the original submission. The statement itself is intended to be a hypothesis that is justified in the remainder of Section 2 (and we have now stated as such in that line).

As for the justification itself, this lies in the comparative timescales of flow to fracture, which is a general statement that can be applied to any damage model. Any such damage model can be evaluated using Equation 6 to determine what the ratio of timescales is. However, we acknowledge that, since Section 2 had focused only on using the Pralong and Funk 2005 version of the damage model to evaluate the validity of the hypothesis, this was not obvious. In this revision, we have added a new section 2.5: Reconciling the Diagnostic Damage Model with Other Damage Models in 2D, which we believe makes this point stronger for two reasons. First, it uses the 2D MISMIP+ geometry and model setup, rather than the simpler flowline model. Second, it directly compares the result of three different damage models: the diagnostic

model proposed in this study, the Pralong and Funk 2005 model, and the Sun et al. 2017 model. We show (see Figure 1 in this response) simulations (with no changes to melt or surface accumulation) that all three models, when set up in a consistent way, produce approximately the same behavior and damage fields.

Importantly, we make the argument that this diagnostic damage model reconciles the two common approaches for modeling damage in ice sheets: the power-law source term of Pralong and Funk 2005 [11] and the Nye Zero Stress approach of Sun et al. 2017 [12] (and many other studies). In this new section, we explain that the diagnostic damage model approximates the Pralong and Funk model when their damage rate factor $B$ is sufficiently large. We also explain, which we did not do well enough in the original submission, that when using experimentally-constrained damage parameters in the Pralong and Funk model, the accumulation of damage is sufficiently large for the diagnostic model to approximate the full transient model. We also show that the diagnostic damage model produces the same behavior as the model of Sun et al. in a certain configuration: that is, when $\sigma_t = 0$ and the stress criterion is the maximum principal stress, which are fundamental assumptions underlying the Nye Zero Stress model. Notably in Figure 1, a cross-glacier rift does not form, as it does in the main text. This is because these simulations do not include any basal melt forcing.

In this way, the diagnostic damage model can be thought of as a more general approach to the Nye Zero Stress approximation, in which we generalize the assumption for any $\sigma_t$ and any stress criterion (and later in the paper we explore the implications of these choices). Finally, we also argue, but don't directly show, that the diagnostic damage model also reconciles the strain-rate-based approach of Albrecht et al. 2012, 2014 [2, 1], in the limit where their damage accumulation factor $\gamma \gg 1$.

We address the specific comment that the results shown in Sun et al. 2017 appear to have less damage than is shown in this study below, where the reviewer specifically makes this comment.

Line 135: could you precise if you use a linear or a non-linear (typically $m = 3$) Weertman friction law. I could not find the information in the supplementary material.

We've now specified that we use a nonlinear Weertman sliding law with $m = 3$.

Line 174: you mention $\sigma_t = 0.02$ MPa as an arbitrary value. Other studies use much higher values (at least about 0.1 MPa) (e.g., Krug et al. 2014; Sun et al., 2017). For Krug et al. (2014), these values have been calibrated with observations of calving rate. Although I understand that your model only aims to simulate damage (without going to calving or riging), I think that the choice of $\sigma_t$ is very important and will largely affects the result. Low $\sigma_t$ allows to easily damage the ice in many regions, and due to the term $(1 - D)^{-k}$ initiated damage leads to even more damage at the next timestep. It also depends on the value of $B$ in Eq. (8).

We've addressed this in more detail above. For the purposes of the flowline model, this value is set low enough to produce damage in a model that does not produce such high stresses as are common in natural systems. Since $\sigma_t$ does not affect the hypothesis of timescales, we do not believe this affects these results. They do, however, affect the significance of damage to mass loss, as explored in Section 3, and there we use more realistic values of $\sigma_t$ and explore the effect of uncertainty in this parameter. We have added text to improve the clarity of this point. We will also rerun Figure 2 for a different $\sigma_t$ value to show that it produces largely the same result.

Lines 216?224 and general statement about Section 3: I think that the limitation of your hypothesis is visible in your supplementary material where Damage seems particularly high and potentially overestimated. For example, your damage is much higher than Sun et al. (2017) but it might also be due to the difference in the criterion used to calculate the damage source () might also present biases due to other limitations in the physical model and its numerical implementation).

[Figure]

Figure 1: **Comparison between 2D transient and diagnostic damage models:** We set up a 2D model geometry as prescribed by the MISMIP+ configuration [3] and run this from a steady state (c-e) without climate forcing with couplings to three different damage models: the model of [11], the model of [12], and the diagnostic damage model proposed in this study. We will refer to the models of [11] and [12] as "full" models. We show mass loss as estimated from each of these three models (a) and the errors between the full models and the diagnostic damage model (b). The error is reported as the mass loss from the full model coupling minus the mass loss from the diagnostic damage model coupling, scaled by the total mass loss from the full model coupling. We show the final damage fields after 100 years for each of the three model couplings (f-h). The red line denotes the grounding line position.

This is a great point and one that we neglected to explain in detail in the original submission. It is true that the diagnostic damage model produces more damage than that shown in Sun et al. 2017 [12]. However, it is not a deficiency in the diagnostic damage model but rather a slight difference in the physics underlying both models. In the new Section 2.5, Reconciling the Diagnostic Damage Model with Other Damage Models in 2D, we will show that the diagnostic damage model and the model of Sun et al. can be derived as the same result, but the model of Sun et al. follows the Nye Zero Stress approximation assumptions of $\sigma_t$ = 0 MPa and a stress criterion of the maximum principal stress (see Figure 1).

However, it's absolutely true that our implementation of the Sun et al. model still produces more damage than is shown in the original paper. This is now explained Section 3.1: Impact of damage production and evolution (lines 455-460). Both the diagnostic damage model and the model of Sun et al. represent damage accumulation with depth as occurring due to stress opening up cracks and ice pressure counteracting damage accumulation. However, [12] defines ice pressure counteracting damage accumulation as $p = \rho_i g z$, as the overburden pressure. This is consistent with previous studies (e.g. [4, 10]). However, our calculation of stress includes the horizontal normal stresses, such that $p = \rho_i g z - \tau_{11} - \tau_{22}$. This ultimately produces less pressure counteracting damage accumulation and, therefore, more damage. Therefore, this difference is not due to the form of the diagnostic damage model but just the physical parameterization of ice pressure. We've made this explicit in this revision.

You allow for your model to reach values of $D$ up to 0.99 to avoid null denominator in the effective stress model and numerical stability and convergence issues. However, is $D$ = 0.99 a realistic value for CDM. While $D < 1$ is a numerical condition to avoid infinite ice fluidity, I think that for too high damage values, especially when over large areas and deep into the column, the CDM really shows its limitations as we continue to simulate something that is far from being continuous as continuous. Later you mention in Section 3 that you set up $D_{max} \sim 0.8$. Can you precise if this is only for the MISMIP+ experiment and that you use $D_{max}$ = 0.99 in the previous experiments?

Yes, $D_{\max}$ = 0.99 for the flowline model and $D_{\max}$ = 0.8 for the 2D simulation to ensure convergence with the numerical solver. This is now stated more clearly for the flowline model. As for what a physical value of $D_{\max}$ is, if we assume that $D$, in this case, is really representing the loss of stress-bearing ability of a representative volume of ice, then in theory it is true that $D$ can reach values of 1 (or very close to 1) as the ice loses its ability to bear stress. Whether this is true for ice is not well-constrained, as far as we know. There are studies that suggest that ice should fail in a brittle manner at lower values of $D$, and some previous studies have parameterized this as a "critical damage" [5]. The specific relationship between fracturing and viscosity, however, has not been widely studied in ice (though is a direction for future work). However, this relationship between stress and the damage variable has a theoretical foundation in materials science principles and is widely adopted in damage studies in glaciology, so without another framework we have chosen to use the most widely-agreed-upon framework.

Line 230: I don't really understand the point of calculating the percentage difference in grounding line position ager 1 year for different $\delta$. You then check the same error after a longer time (e.g., $10^2 - 10^4$ years) for which your diagnostic model is supposed to be more valid, which to me is a much better way to look at the ?error? of the diagnostic model.

This is a great point. We have rerun Figure 2 to 1000 years, rather than 1 year; see Figure 2 in this response.

Line 266: you mention healing as a form of damage sink in the advection equation. You might consider mentioning that the right end side of Eq. (1) is a ?damage source/sink? when you present the equation.

The text does mention that the right hand side of Equation 1 can be either a source or sink "$f$ is the damage evolution function describing the rate of change of damage due to deterioration or healing". We

[Figure]

Figure 2: **Timescales for which the diagnostic damage model is applicable:** Parameter space of fracture timescale and advective timescale for (a) nondimensional parameter $\delta$, (b) the error in the grounding line position using the diagnostic damage solver (the difference between grounding line position found from coupling the flowline model with the transient [11] damage model and the grounding line position found from coupling the flowline model with the diagnostic damage model, scaled by the total grounding line change in the transient model) after a simulation time of 1000 years. The red box represents likely range of values of advective and fracture timescales based on geometry of ice streams and the physics of fracture, and the green dot denotes the parameters used in the flowline model.

acknowledge, however, that this terminology wasn't consistent and that we continued to use "source term" for the remainder of the study. We have replaced all uses of "source term" with either "source/sink" or "damage evolution function" more broadly.

Line 286: Could you precise the resolution of the mesh in the vicinity of the grounding line?

This is now specified.

Line 310: I would add a comma: ?We note that in 3D, cracks are ...?

Done.

Lines 310?315 (this could actually be a third main concern): The model is Shallow Shelf (SSA) and therefore computes horizontal velocities considered constant over the ice column (plug flow). You can therefore assume null vertical velocities but this would greatly limit damage deep into the column (i.e., no vertical advection of the crevasses). You might also neglect ?significant? vertical stress components, potentially leading to inaccurate stress calculations.

Assuming mass conservation, you can recompute vertical velocities (based on surface and basal accumulation/melt and some assumption on the distribution) but you would need to assume a vertical distribution. Could you give more detail here?

Since a lot of the damage creates close to the grounding line where the SSA solution and the resulting stress computation is more prone to errors, what impact do you think the Stokes approximation has on the damage solution (you mention the role of longitudinal, lateral and shear deformation on crack closure later but not on the creation of crevasses/damage them/itself)? This would be interesting to discuss.

Thank you for this point; the ice flow model icepack does solve the SSA equations and therefore neglects some vertical stress components and the vertical advection of damage. Given how uncertain rates of vertical advection in ice sheets are, we believe that including vertical advection may add even more uncertainty than including it would. We agree that vertical velocities may be calculated from accumulation and basal melt, but the assumption of vertical distribution is a significant one and we are not aware of good constraints on this. Furthermore, the effect of ice overburden pressure may be to close cracks that advect into compressive stress states, and the mechanisms of crack opening and crack healing in compressive states is not well understood. Since most of the damage accumulation occurs on the ice shelves, where the SSA is likely to be most applicable, we believe the most significant effect may be to reduce the amount of damage on grounded ice by not accounting for vertical shear as a potential crack opening mechanism. We have discussed these assumptions in a new paragraph in the Discussion section.

Line 336: ?SUpplement? → ?Supplement?

Done.

Line 346: I suggest ?... and run the model for 100 years in two simulations?. instead of ?... and run the simulation for 100 model years in two simulations? .

Done.

Line 515: The blurriness of sharp cracks is also due to stabilization techniques for the advection equation. Stabilization techniques often rely on artificial diffusion, even for small timesteps and or small CFL numbers.

In this case, the blurriness of sharp cracks would likely exist even without any stabilization techniques due to the longer timestep size and significant increases in damage with each timestep. This will be explained more clearly in the revision.

Line 523: From this statement, I understand that there is no advection of damage in the column except

for the fact that if the ice base is melted, a larger part of the column could be affected by damage? As I said in a previous comment, a part of the ?vertical advection? of the damage could therefore be due to a numerical vertical diffusion of the solution. I think this could be better discussed here.

This is correct, there is no vertical advection of damage in this model. There should not be any numerical vertical diffusion of damage in the model, as there are no "numerics" associated with the diagnostic damage model in 3D. It's not obvious whether, if we incorporate vertical stresses, more of the ice column would be affected by damage. As mentioned in the reply to the previous comment, this would be a question of whether, once those cracks advect into a net compressive stress states, they would remain or heal. The assumption of SSA is now explored in the Discussion section.

**References**

[1] T. Albrecht and A. Levermann. Fracture-induced softening for large-scale ice dynamics. *The Cryosphere*, 8(2):587–605, April 2014.

[2] Torsten Albrecht and Anders Levermann. Fracture field for large-scale ice dynamics. *Journal of Glaciology*, 58(207):165–176, 2012.

[3] Xylar S. Asay-Davis, Stephen L. Cornford, Gal Durand, Benjamin K. Galton-Fenzi, Rupert M. Gladstone, G. Hilmar Gudmundsson, Tore Hattermann, David M. Holland, Denise Holland, Paul R. Holland, Daniel F. Martin, Pierre Mathiot, Frank Pattyn, and Hlne Seroussi. Experimental design for three interrelated marine ice sheet and ocean model intercomparison projects: MISMIP v. 3 (MISMIP +), ISOMIP v. 2 (ISOMIP +) and MISOMIP v. 1 (MISOMIP1). *Geoscientific Model Development*, 9(7):2471–2497, July 2016.

[4] Douglas I. Benn, Nicholas R.J. Hulton, and Ruth H. Mottram. Calving laws, sliding laws and the stability of tidewater glaciers. *Annals of Glaciology*, 46:123–130, 2007.

[5] Ravindra Duddu, Stephen Jimnez, and Jeremy Bassis. A non-local continuum poro-damage mechanics model for hydrofracturing of surface crevasses in grounded glaciers. *Journal of Glaciology*, 66(257):415–429, June 2020.

[6] Alex Huth, Ravindra Duddu, and Ben Smith. A Generalized Interpolation Material Point Method for Shallow Ice Shelves. 1: Shallow Shelf Approximation and Ice Thickness Evolution. *Journal of Advances in Modeling Earth Systems*, 13(8), August 2021.

[7] Alex Huth, Ravindra Duddu, and Ben Smith. A Generalized Interpolation Material Point Method for Shallow Ice Shelves. 2: Anisotropic Nonlocal Damage Mechanics and Rift Propagation. *Journal of Advances in Modeling Earth Systems*, 13(8), August 2021.

[8] Alex Huth, Ravindra Duddu, Benjamin Smith, and Olga Sergienko. Simulating the processes controlling ice-shelf rift paths using damage mechanics. *Journal of Glaciology*, pages 1–14, September 2023.

[9] O Mahrenholtz and Z Wu. Determination of creep damage parameters for polycrystalline ice. In *Third International Conference on Ice Technology, Advances in Ice Technology*, Massachusetts Institute of Technology, Cambridge, MA, 1992.

[10] F.M. Nick, C.J. Van Der Veen, A. Vieli, and D.I. Benn. A physically based calving model applied to marine outlet glaciers and implications for the glacier dynamics. *Journal of Glaciology*, 56(199):781–794, 2010.

[11] A. Pralong and M. Funk. Dynamic damage model of crevasse opening and application to glacier calving. *Journal of Geophysical Research*, 110(B1):B01309, 2005.

[12] Sainan Sun, Stephen L. Cornford, John C. Moore, Rupert Gladstone, and Liyun Zhao. Ice shelf fracture parameterization in an ice sheet model. *The Cryosphere*, 11(6):2543–2554, November 2017.

---

## Author Comment (AC2)

The comments from the reviewers are in black font and our responses are in blue font.

As the other reviewer mentions, I think the authors should be clear on the uncertainties within their estimates of the advection and production timescales, and how does this propogate through to $\delta$. What is the maximum and minimum value that $\delta$ can take given these uncertainties on the stress threshold, glacier length etc.? Where might this assumption not be valid?

This is a great point. We have added text in the revision to provide clarity about this point. $\delta$ depends on both the advective timescale (which itself depends on the characteristic length scale and velocity scale of the glacier) and the fracture timescale (which is set by the damage model; in the case of the Pralong and Funk model, this depends on the stress scale, the damage rate factor $B$, and the damage exponent $r$). It does not depend on the stress threshold, which only sets the spatial locations for which fractures will begin to accumulate. The fracture timescale parameters we use are taken from experimental results and thus are calibrated for ice.

The range of values that $\delta$ may be in ice are shown in the shaded regions of Figure 2 (which is now redone to show results for 1000 year simulations, rather than 1 year simulations, based on a suggestion by the other reviewer). As is pointed out in this review, this depends on things like the glacier length scale, which can vary between glaciers. We intended this to be a broad range that could encompass many different ice sheet glacier configurations, though we have set up this theory so that the advective and fracture timescales can be individually calculated for any simulation a modeler wishes to run to determine whether their setup allows for the diagnostic damage model. This assumption would not be valid, for instance, for representing rapid fracture processes on a short timescale (e.g. rifting and calving events), as demonstrated by the $\delta$ parameter space.

The supplement is significant, and I think it would be helpful to move some of it into the main text. In particular, I think Fig S2 and the accompanying text would be helpful to further show that this diagnostic model is still valid in less idealised configurations.

We agree with this point. We have moved the comparison between the transient and diagnostic models in a 2D (less idealized) configuration into the main text, in a new section 2.5: Reconciling the Diagnostic Damage Model with Other Damage Models in 2D. This section presents comparisons between the diagnostic damage model and two other commonly-used damage models: Pralong and Funk 2005 model and Sun et al. 2017 model (see Figure 1 of this response). We show that, with consistent model setups, the diagnostic damage model replicates both of these model results very well and produces very similar ice mass loss estimates.

The rest of the supplement is primarily extra parameter sensitivity studies (e.g. other melt scenarios, mesh sizes and timesteps, $D_{\mathrm{max}}$ values, choice of pressure calculation) and other experiments for comparison to the MISMIP+ experiments (Ice 1ra, a simulation out to 500 years rather than 100 years) and therefore, we believe, aren't needed in the main text for the ultimate takeaways of the paper.

Line 324: Am I correct in thinking that limiting the value of Dmax effectively limits the maximum softening caused by damage? Such that Dmax = 0.5 gives a max softening of 8 etc.? If so this may be helpful to explain in the text.

Yes this is correct and is now formally stated.

Fig 5: The label for (b) (i) ?Damage field after 100 years? is directly over a subplot for t=0 years: this label should be changed for part (b) of the figure.

Changed to "Damage fields".

Supplement Line 119: WhenIf

[Figure]

Figure 1: **Comparison between 2D transient and diagnostic damage models:** We set up a 2D model geometry as prescribed by the MISMIP+ configuration [1] and run this from a steady state (c-e) without climate forcing with couplings to three different damage models: the model of [2], the model of [3], and the diagnostic damage model proposed in this study. We will refer to the models of [2] and [3] as "full" models. We show mass loss as estimated from each of these three models (a) and the errors between the full models and the diagnostic damage model (b). The error is reported as the mass loss from the full model coupling minus the mass loss from the diagnostic damage model coupling, scaled by the total mass loss from the full model coupling. We show the final damage fields after 100 years for each of the three model couplings (f-h). The red line denotes the grounding line position.

Fixed.

**References**

[1] Xylar S. Asay-Davis, Stephen L. Cornford, Gal Durand, Benjamin K. Galton-Fenzi, Rupert M. Glad-stone, G. Hilmar Gudmundsson, Tore Hattermann, David M. Holland, Denise Holland, Paul R. Holland, Daniel F. Martin, Pierre Mathiot, Frank Pattyn, and Hlne Seroussi. Experimental design for three interrelated marine ice sheet and ocean model intercomparison projects: MISMIP v. 3 (MISMIP +), ISOMIP v. 2 (ISOMIP +) and MISOMIP v. 1 (MISOMIP1). *Geoscientific Model Development*, 9(7):2471–2497, July 2016.

[2] A. Pralong and M. Funk. Dynamic damage model of crevasse opening and application to glacier calving. *Journal of Geophysical Research*, 110(B1):B01309, 2005.

[3] Sainan Sun, Stephen L. Cornford, John C. Moore, Rupert Gladstone, and Liyun Zhao. Ice shelf fracture parameterization in an ice sheet model. *The Cryosphere*, 11(6):2543–2554, November 2017.

---

## Author Response (AR1)

We thank both reviewers and the editor for their insightful comments, which we believe strengthened this paper significantly. Below please find our responses to the reviewers' comments. The comments from the reviewers are in black font and our responses are in blue font. Line numbers correspond to the tracked-changes version of the main text.

**1   Review 1**

**1.1   Hypothesis for the diagnostic damage model, i.e., fractures accumulate much faster than they advect.**

The choice of stress threshold in the model must have a significant impact on the results, as it directly influences where and how easily damage initiates. In the first experiment, the authors use $\sigma_t = 0.02$ MPa, a very low threshold that contrasts with higher values used in other studies (e.g., Krug et al., 2014; Sun et al., 2017), sometimes calibrated to match observed calving rates. While I understand that the model?s focus is on damage accumulation rather than explicit calving, this low threshold promotes rapid damage development across broader ice regions than higher threshold values. This, combined with the feedback effect of damage on the source term (i.e., $f = f((1 - D)^{-k})$), means that initial damage quickly propagates in subsequent timesteps. Given these sensitivities, a discussion of the implications for using a low $\sigma_t$, as well as the role of the damage rate factor $B$, would help better justifying the hypothesis.

Without this, as a reader, I feel like the value picked for $\sigma_t$ and $B$ (I could not find the value) affect the realism of the hypothesis $\delta \gg 1$, leading to fast damage creation rate. It makes it look like the experiment is built to match the hypothesis. While I do not see any argument to revoke the hypothesis, I do not think either that the results validate the hypothesis – although later experiments seem to show little effect of $\sigma_t$ which could be specific to the experiments? The sentence "This agrees with the theory that where $\delta \gg 1$, fractures accumulate much more rapidly than they are advected away by ice flow" is therefore a bit of a stretch to me. Since this hypothesis and the first experiment condition the definition of $D_{acc}$ . I think that these limitations should be better discussed

We thank the reviewer for this point and acknowledge that some of this discussion was not clear enough in the initial submission. We agree that the value of $\sigma_t$ is quite a bit lower than we would expect in natural ice sheets. This value was chosen because of the simplicity of the flowline model; it produces generally very low stresses, requiring a low stress threshold in order to produce fracture. Since we wanted to study the importance of the rate of damage accumulation, we had to pick a stress threshold that would produce some amount of damage. This is now stated explicitly in lines 187-191.

However, we don't believe this value being low should affect the results shown in Figure 2. While we agree that the choice of the stress threshold significantly affects the spatial extent and intensity of damage produced, the stress threshold does not play a substantial role in the rate of damage production; this is modulated predominantly by, in the case of the [12] model, the two fracture parameters $B$ and $r$, as well as the characteristic stress scale (which depends, in the flowline model, on ice rheology, the velocity scale, and the length scale). This is stated more clearly now in lines 250-260. While this isn't necessarily obvious from the equations, we have also now rerun Figure 2 for a different strses threshold value ($\sigma_t = 0.05$ MPa), included in the supplement (Figure S1), to show that it largely shows the same results as in Figure 2 (and this is mentioned in lines 259-260). Due to time and computational requirements, the supplement version of the figure is at lower resolution but produces the same result. As described in more detail below, we also now do a direct comparison between the diagnostic damage model and a full transient model in 2D with more realistic parameter values (Section 2.5; lines 295-393), to show that these results are robust to

the idealized nature of the flowline model and any variations in the parameter values from realistic values.

Finally, to the reviewer's point about discussing the effect of $B$ and $r$, the parameter study in Figure 2 does consider the effect of these parameters on this hypothesis. In Figure 2, we show in which regions of advective and fracture timescale space our hypothesis (that fractures accumulate much faster than the rate of flow) is valid. The fracture timescale is, effectively, a combination of the parameters $B$ and $r$, as defined in the manuscript as:

$$[t_f] = B[\tilde{\sigma}]^r \tag{1}$$

Therefore, varying the fracture timescale effectively varies the fracture parameters, and Figure 2 shows that the hypothesis that fracture accumulation is much faster than flow advection does not apply for low values of $B$ and $r$. We state this more clearly now in lines 127-129, 255-257. We present Fig. 2 in fracture timescale space, rather than $B$ space, to make it generally applicable to many different damage models (that is, someone could calculate the fracture timescale of a different damage model and determine if this hypothesis is valid using Fig.2).

The applicability, however, to a specific problem does require determining the fracture and flow timescales. Doing this, barring any intrinsic or physical understanding of fracture timescales, requires us to assume some damage model and calculate the timescale to verify where it falls in $\delta$ space. We state this clearly now in lines 250-260. This is where we, for illustrative purposes, apply the Pralong and Funk model for comparison. The specific values of $B$ and $r$ are those widely applied in other glaciology studies building on the Pralong and Funk model [12, 6] and were originally constrained by laboratory experiments [9] and thus were not chosen to fit this particular problem. We are more careful in this revision to state this explicitly in lines 171 and 302-204.

**1.2   Unsymmetrical damage for a symmetrical geometry and numerical artifact**

I would suggest the authors discuss whether such factors could contribute to any asymmetry observed in their model results and clarify the type of stabilization methods employed, as well as any expected artifacts. Addressing these potential sources of numerical asymmetry (and diffusion) would strengthen numerical aspect of the study.

Since the MISMIP+ configuration is symmetrical along the central flowline, any asymmetry in the modeled damage field (or other model output) may indicate numerical artifacts rather than genuine physical behavior. Artifacts of this nature often arise from issues like model stability (e.g., CFL condition) or other challenges in the advection scheme, particularly in handling diffusion-free processes like damage. In these cases, artificial diffusion or other stabilization techniques are often used, which can introduce numerical artifacts.

This is a great point, it is correct that these asymmetries do not arise from the model configuration, given that both the MISMIP+ geometry and the forcing are symmetric. While there are no stabilization techniques imposed in the solver, this asymmetry could be due to a number of factors. We agree that numerical diffusion does occur in Eulerian advection schemes, and small amounts of numerical diffusion could explain these asymmetries. This could also arise from physical symmetry breaking, in which asymmetric patterns can arise in systems that contain symmetric equations and forcing due to, for example, small errors from numerical approximations amplifying. Given that reductions in the timestep do not have significant effects on the solutions (Supplement Fig. S4), the solution appears stable towards numerics and thus this could be evidence for physical symmetry breaking explaining these asymmetries. This could also arise due to asymmetries in the mesh itself, which can be seen in some of the grounding line profiles. While we don't

believe this affects the scientific conclusions of the paper, we agree that it is important to explain these artifacts. We've added a discussion on this in lines 520-526 when discussing the MISMIP+ results. We also discuss methods for addressing these numerical diffusion issue, such as the Material Point Method employed by previous papers [5, 6, 7].

**1.3    Specific Comments**

Line 17: I found this first sentence a little strange, I guess that the authors refer to 'viscous ice flow' as opposed to 'elasto-brittle calving'. Maybe consider rephrasing this.

This sentence is intended to describe the viscous response of ice sheets, rather than brittle calving. We have simplified this to just say "Ice flow" (line 17).

Line 30: the term 'instability' is not really clear here. I would detail a little more the processes it refers to.

We've specified more clearly what we mean in that line (line 30).

Line 40: I would expand the ice sheet model timescale to as low as $\sim 10^{-1}$ year. More and more models are interested in seasonal changes (without going to visco-elastic models interested in tidal effect, ...).

Done.

Line 60: I would change 'accumulated fractures' for 'accumulated damage' since CDM was used to enhance damage, until damage was deep enough to trigger a fracture over the entire column with LEFM.

Done.

Line 72: switch the order of the two citations

Done.

Line 102: 'We evaluate the applicability'

Done.

Line 106: I think that TC writes equations as this: Eq. (1). Think about correcting this for other references to your equations

Done.

Equation 5: You therefore assume that $[t_a] = [t]$ to simplify your first term but isn't $[t]$ the resultant of the advective ($[t_a]$) and the fracture ($[t_f]$) timescale? I guess you assume $[t_f] \ll [t_a]$ to make the simplification but this relies on the hypothesis you do after. I might be missing something here.

The timescale $[t]$ that we had defined in the first submission was describing the timescale of the flow problem, rather than the fracture problem, and therefore was by definition the advective timescale. In this revision, we have defined the timescale of the flow problem up front and removed any use of $[t]$, to avoid confusion (lines 115-117).

Line 130: how do you justify this hypothesis for a typical damage model?

Thanks for this point; we acknowledge that this should have been made far more explicit in the original submission. The statement itself is intended to be a hypothesis that is justified in the remainder of Section 2 (and we have now stated as such in line 134).

As for the justification itself, this lies in the comparative timescales of flow to fracture, which is a general statement that can be applied to any damage model. Any such damage model can be evaluated using Equation 6 to determine what the ratio of timescales is. However, we acknowledge that, since Section 2

had focused only on using the Pralong and Funk (2005) version of the damage model to evaluate the validity of the hypothesis, this was not obvious. In this revision, we have added a new section 2.5: Reconciling the Diagnostic Damage Model with Other Damage Models in 2D (lines 295-393), which we believe makes this point stronger for two reasons. First, it uses the 2D MISMIP+ geometry and model setup, rather than the simpler flowline model. Second, it directly compares the result of three different damage models: the diagnostic model proposed in this study, the Pralong and Funk 2005 model, and the Nye zero stress model. We show (see Figure 1 in this response) simulations (with no changes to melt or surface accumulation) that all three models, when set up in a consistent way, produce approximately the same mass loss and damage fields.

Importantly, we make the argument that this diagnostic damage model reconciles the two common approaches for modeling damage in ice sheets: the power-law source term of Pralong and Funk (2005) [12] and the Nye zero stress approach of Sun et al. (2017) [13] (and many other studies). In this new section, we explain that the diagnostic damage model approximates the Pralong and Funk model when their damage rate factor $B$ is sufficiently large. We also explain, which we did not do well enough in the original submission, that when using experimentally-constrained damage parameters in the Pralong and Funk model, the accumulation of damage is sufficiently large for the diagnostic model to approximate the full transient model. We also show that the diagnostic damage model produces the same behavior as the Nye zero stress model (similar to that applied in the Sun et al. 2017 study [13]) in a certain configuration: that is, when $\sigma_t = 0$ and the stress criterion is the maximum principal stress, which are fundamental assumptions underlying the Nye zero stress model. Notably, in Figure 1, a cross-glacier rift does not form, as it does in the main text. This is because these simulations do not include any basal melt forcing.

In this way, the diagnostic damage model can be thought of as a more general approach to the Nye zero stress approximation, in which we generalize the assumption for any $\sigma_t$ and any stress criterion (and later in the paper we explore the implications of these choices). Finally, we also argue, but don't directly show, that the diagnostic damage model also reconciles the strain-rate-based approach of Albrecht et al. (2012, 2014) [2, 1], in the limit where their damage accumulation factor $\gamma \gg 1$.

We address the specific comment that the results shown in Sun et al. (2017) appear to have less damage than is shown in this study below, where the reviewer specifically makes this comment.

Line 135: could you be precise if you use a linear or a non-linear (typically $m = 3$) Weertman friction law. I could not find the information in the supplementary material.

We've now specified that we use a nonlinear Weertman sliding law with $m = 3$ (line 140).

Line 174: you mention $\sigma_t = 0.02$ MPa as an arbitrary value. Other studies use much higher values (at least about 0.1 MPa) (e.g., Krug et al. 2014; Sun et al., 2017). For Krug et al. (2014), these values have been calibrated with observations of calving rate. Although I understand that your model only aims to simulate damage (without going to calving or rifting), I think that the choice of $\sigma_t$ is very important and will largely affects the result. Low $\sigma_t$ allows to easily damage the ice in many regions, and due to the term $(1 - D)^{-k}$ initiated damage leads to even more damage at the next timestep. It also depends on the value of $B$ in Eq. (8).

We've addressed this in more detail above. For the purposes of the flowline model, this value is set low enough to produce damage in a model that does not produce such high stresses as are common in natural systems. Since $\sigma_t$ does not significantly affect the hypothesis of timescales, we do not believe this affects these results. They do, however, affect the significance of damage to mass loss, as explored in Section 3, and there we use more realistic values of $\sigma_t$ and explore the effect of uncertainty in this parameter. We have added text to improve the clarity of this point (127-129, 187-191, 255-257). We have also rerun Figure 2 for a different $\sigma_t$ value to show that it produces largely the same result (now in the Supplement Fig. S1).

[Figure]

Figure 1: **Comparison between 2D transient and diagnostic damage models:** We set up a 2D model geometry as prescribed by the MISMIP+ configuration [3] and run this from a steady state (c-e) without climate forcing with couplings to three different damage models: the model of [12], the model of [13], and the diagnostic damage model proposed in this study. We will refer to the models of [12] and [13] as "full" models. We show mass loss as estimated from each of these three models (a) and the errors between the full models and the diagnostic damage model (b). The error is reported as the mass loss from the full model coupling minus the mass loss from the diagnostic damage model coupling, scaled by the total mass loss from the full model coupling. We show the final damage fields after 100 years for each of the three model couplings (f-h). The red line denotes the grounding line position.

Lines 216–224 and general statement about Section 3: I think that the limitation of your hypothesis is visible in your supplementary material where damage seems particularly high and potentially overestimated. For example, your damage is much higher than Sun et al. (2017) but it might also be due to the difference in the criterion used to calculate the damage source (might also present biases due to other limitations in the physical model and its numerical implementation).

This is a great point and one that we neglected to explain in detail in the original submission. It is true that the diagnostic damage model produces more damage than that shown in Sun et al. 2017 [13]. However, it is not a deficiency in the diagnostic damage model but rather a slight difference in the physics underlying both models. In the new Section 2.5, Reconciling the Diagnostic Damage Model with Other Damage Models in 2D (lines 295-393), we will show that the diagnostic damage model and the Nye zero stress model (similar to that applied in [13]) can be derived as the same result, but the Nye zero stress approximation relies on the assumptions of $\sigma_t = 0$ MPa and a stress criterion of the maximum principal stress (see Figure 1 of this response).

However, it's true that our implementation of the Nye zero stress model when applied to the MISMIP+ experiment (with basal melting) still produces more damage than is shown in the original paper of [13], as it would also form a cross-ice shelf rift as the diagnostic damage model does. This is now explained Section 3.1: Impact of damage production and evolution (lines 455-460). Both the diagnostic damage model and the model of Sun et al. represent damage accumulation with depth as occurring due to stress opening up cracks and ice pressure counteracting damage accumulation. However, [13] defines ice pressure counteracting damage accumulation as $p = \rho_i gz$, as the overburden pressure. This is consistent with previous studies (e.g. [4, 10]). However, our calculation of ice pressure (based on the shallow shelf approximation) includes the horizontal normal stresses, such that $p = \rho_i gz - \tau_{11} - \tau_{22}$ as in [8, 6, 7]. This ultimately produces less pressure counteracting damage accumulation and, therefore, more damage. Therefore, this difference is not due to the form of the diagnostic damage model but just the physical parameterization of ice pressure. We've made this explicit in this revision (lines 471-476).

You allow for your model to reach values of $D$ up to 0.99 to avoid null denominator in the effective stress model and numerical stability and convergence issues. However, is $D = 0.99$ a realistic value for CDM. While $D < 1$ is a numerical condition to avoid infinite ice fluidity, I think that for too high damage values, especially when over large areas and deep into the column, the CDM really shows its limitations as we continue to simulate something that is far from being continuous as continuous. Later you mention in Section 3 that you set up $D_{max} \sim 0.8$. Can you be precise if this is only for the MISMIP+ experiment and that you use $D_{max} = 0.99$ in the previous experiments?

Yes, $D_{\max} = 0.99$ for the flowline model and $D_{\max} = 0.8$ for the 2D simulation to ensure convergence with the numerical solver. This is now stated more clearly for the flowline model in lines 153-154. As for what a physical value of $D_{\max}$ is, if we assume that $D$, in this case, is really representing the loss of stress-bearing ability of a representative volume of ice, then in theory it is true that $D$ can reach values of 1 (or very close to 1) as the ice loses its ability to bear stress. Previous work has shown that $D_{\max} \approx 1$ does well model a full-thickness fracture, provided proper boundary conditions are set, and the same study also shows that a lower $D_{\max}$ value (of $D_{\max} = 0.86$) provides a good approximation and allows for slow rift separation seen in observations [7]. Both cases replicate rifting on the Larsen C ice shelf well [7].

Line 230: I don't really understand the point of calculating the percentage difference in grounding line position after 1 year for different $\delta$. You then check the same error after a longer time (e.g., $10^2 - 10^4$ years) for which your diagnostic model is supposed to be more valid, which to me is a much better way to look at the 'error' of the diagnostic model.

This is a great point. We have rerun Figure 2 to 1000 years, rather than 1 year; see Figure 2 in this

[Figure]

Figure 2: **Timescales for which the diagnostic damage model is applicable:** Parameter space of fracture timescale and advective timescale for (a) nondimensional parameter $\delta$, (b) the error in the grounding line position using the diagnostic damage solver (the difference between grounding line position found from coupling the flowline model with the transient [12] damage model and the grounding line position found from coupling the flowline model with the diagnostic damage model, scaled by the total grounding line change in the transient model) after a simulation time of 1000 years. The red box represents likely range of values of advective and fracture timescales based on geometry of ice streams and the physics of fracture, and the green dot denotes the parameters used in the flowline model.

response.

Line 266: you mention healing as a form of damage sink in the advection equation. You might consider mentioning that the right end side of Eq. (1) is a 'damage source/sink' when you present the equation.

The text does mention that the right hand side of Equation 1 can be either a source or sink "$f$ is the damage evolution function describing the rate of change of damage due to deterioration or healing". We acknowledge, however, that this terminology wasn't consistent and that we continued to use "source term" for the remainder of the study. We have replaced all uses of "source term" with either "source/sink" or "damage evolution function" more broadly.

Line 286: Could you be precise the resolution of the mesh in the vicinity of the grounding line?

This is specified in line 347.

Line 310: I would add a comma: "We note that in 3D, cracks are ..."

Done.

Lines 310–315 (this could actually be a third main concern): The model is Shallow Shelf (SSA) and therefore computes horizontal velocities considered constant over the ice column (plug flow). You can therefore assume null vertical velocities but this would greatly limit damage deep into the column (i.e., no vertical advection of the crevasses). You might also neglect 'significant' vertical stress components, potentially leading to inaccurate stress calculations.

Assuming mass conservation, you can recompute vertical velocities (based on surface and basal accumulation/melt and some assumption on the distribution) but you would need to assume a vertical distribution. Could you give more detail here?

Since a lot of the damage creates close to the grounding line where the SSA solution and the resulting stress computation is more prone to errors, what impact do you think the Stokes approximation has on the damage solution (you mention the role of longitudinal, lateral and shear deformation on crack closure later but not on the creation of crevasses/damage them/itself)? This would be interesting to discuss.

Thank you for this point; the ice flow model icepack does solve the SSA equations and therefore neglects some vertical stress components and the vertical advection of damage. Given how uncertain rates of vertical advection in ice sheets are, we believe that including vertical advection may add even more uncertainty than including it would. We agree that vertical velocities may be calculated from accumulation and basal melt, but the assumption of vertical distribution is a significant one and we are not aware of good constraints on this. Furthermore, the effect of ice overburden pressure may be to close cracks that advect into compressive stress states, and the mechanisms of crack opening and crack healing in compressive states is not well understood. Since most of the damage accumulation occurs on the ice shelves, where the SSA is likely to be most applicable, we believe the most significant effect may be to reduce the amount of damage on grounded ice by not accounting for vertical shear as a potential crack opening mechanism. We have discussed these assumptions in a new paragraph in the Discussion section, in lines 654-661.

Line 336: 'SUpplement' → 'Supplement'

Done.

Line 346: I suggest "... and run the model for 100 years in two simulations". instead of "... and run the simulation for 100 model years in two simulations".

Done.

Line 515: The blurriness of sharp cracks is also due to stabilization techniques for the advection equation. Stabilization techniques often rely on artificial diffusion, even for small timesteps and or small CFL numbers.

In this case, the blurriness of sharp cracks would likely exist even without any stabilization techniques due to the Eulerian continuum description, longer timestep size and significant increases in damage with each timestep. This is now pointed out and explained in lines 520-526.

Line 523: From this statement, I understand that there is no advection of damage in the column except for the fact that if the ice base is melted, a larger part of the column could be affected by damage? As I said in a previous comment, a part of the "vertical advection" of the damage could therefore be due to a numerical vertical diffusion of the solution. I think this could be better discussed here.

This is correct, there is no vertical advection of damage in this model. There should not be any numerical vertical diffusion of damage in the model, as there are no "numerics" associated with the diagnostic damage model in 3D. It's not obvious whether, if we incorporate vertical stresses, more of the ice column would be affected by damage. As mentioned in the reply to the previous comment, this would be a question of whether, once those cracks advect into a net compressive stress states, they would remain or heal. The assumption of SSA is now explored in the Discussion section in lines 654-661.

**2   Review 2**

As the other reviewer mentions, I think the authors should be clear on the uncertainties within their estimates of the advection and production timescales, and how does this propogate through to $\delta$. What is

the maximum and minimum value that $\delta$ can take given these uncertainties on the stress threshold, glacier length etc.? Where might this assumption not be valid?

This is a great point. $\delta$ depends on both the advective timescale (which itself depends on the characteristic length scale and velocity scale of the glacier) and the fracture timescale (which is set by the damage model; in the case of the Pralong and Funk model, this depends on the stress scale, the damage rate factor $B$, and the damage exponent $r$). $\delta$ does not depend strongly on the stress threshold, which predominantly sets the spatial locations for which fractures will begin to accumulate. This is further explained now in lines 127-129, 250-260.

The range of values that $\delta$ may be in ice are shown in the shaded regions of Figure 2 (which is now redone to show results for 1000 year simulations, rather than 1 year simulations, based on a suggestion by the other reviewer). As is pointed out in this review, this depends on things like the glacier length scale, which can vary between glaciers. We intended this to be a broad range that could encompass many different ice sheet glacier configurations, though we have set up this theory so that the advective and fracture timescales can be individually calculated for any simulation a modeler wishes to run to determine whether their setup allows for the diagnostic damage model. This is now clearly stated in lines 250-260. As an example for when this diagnostic model would not be applicable, this model could not be used to represent rapid fracture processes on a short timescale (e.g. rifting and calving events), as demonstrated by the $\delta$ parameter space.

We also have expanded sections to justify this assumption for our application of the effect of fracturing on long timescale flow. The fracture timescale parameters we use are taken from experimental results and thus are calibrated for ice (and this is clearly stated now in lines 171 and 302-304). To further illuminate this point, we have repeated Fig. 2 using a different stress threshold value in the Supplement (Fig. S2). Furthermore, as described in more detail below, we have expanded the comparison between the diagnostic damage model and "full" transient damage models to include a two-dimensional comparison that is set up to better approximate natural glaciers.

The supplement is significant, and I think it would be helpful to move some of it into the main text. In particular, I think Fig S2 and the accompanying text would be helpful to further show that this diagnostic model is still valid in less idealised configurations.

We agree with this point. We have moved the comparison between the transient and diagnostic models in a 2D (less idealized) configuration into the main text, in a new section 2.5: Reconciling the Diagnostic Damage Model with Other Damage Models in 2D (lines 295-393). This section presents comparisons between the diagnostic damage model and two other commonly-used damage models: a creep damage model [12] and the Nye zero stress model [11] (see Figure 1 of this response). We show that, with consistent model setups, the diagnostic damage model replicates both of these model results very well and produces very similar ice mass loss estimates.

The rest of the supplement is primarily extra parameter sensitivity studies (e.g. other melt scenarios, mesh sizes and timesteps, $D_{\max}$ values, choice of pressure calculation) and other experiments for comparison to the MISMIP+ experiments (Ice 1ra, a simulation out to 500 years rather than 100 years) and therefore, we believe, aren't needed in the main text for the ultimate takeaways of the paper.

Line 324: Am I correct in thinking that limiting the value of Dmax effectively limits the maximum softening caused by damage? Such that Dmax = 0.5 gives a max softening of 8 etc.? If so this may be helpful to explain in the text.

Yes this is correct and is now formally stated in line 366-368.

Fig 5: The label for (b) (i) "Damage field after 100 years" is directly over a subplot for t=0 years: this

label should be changed for part (b) of the figure.

Changed to "Damage fields".

Supplement Line 119: WhenIf

Fixed.

**References**

[1] T. Albrecht and A. Levermann. Fracture-induced softening for large-scale ice dynamics. *The Cryosphere*, 8(2):587–605, April 2014.

[2] Torsten Albrecht and Anders Levermann. Fracture field for large-scale ice dynamics. *Journal of Glaciology*, 58(207):165–176, 2012.

[3] Xylar S. Asay-Davis, Stephen L. Cornford, Gal Durand, Benjamin K. Galton-Fenzi, Rupert M. Gladstone, G. Hilmar Gudmundsson, Tore Hattermann, David M. Holland, Denise Holland, Paul R. Holland, Daniel F. Martin, Pierre Mathiot, Frank Pattyn, and Hlne Seroussi. Experimental design for three interrelated marine ice sheet and ocean model intercomparison projects: MISMIP v. 3 (MISMIP +), ISOMIP v. 2 (ISOMIP +) and MISOMIP v. 1 (MISOMIP1). *Geoscientific Model Development*, 9(7):2471–2497, July 2016.

[4] Douglas I. Benn, Nicholas R.J. Hulton, and Ruth H. Mottram. Calving laws, sliding laws and the stability of tidewater glaciers. *Annals of Glaciology*, 46:123–130, 2007.

[5] Alex Huth, Ravindra Duddu, and Ben Smith. A Generalized Interpolation Material Point Method for Shallow Ice Shelves. 1: Shallow Shelf Approximation and Ice Thickness Evolution. *Journal of Advances in Modeling Earth Systems*, 13(8), August 2021.

[6] Alex Huth, Ravindra Duddu, and Ben Smith. A Generalized Interpolation Material Point Method for Shallow Ice Shelves. 2: Anisotropic Nonlocal Damage Mechanics and Rift Propagation. *Journal of Advances in Modeling Earth Systems*, 13(8), August 2021.

[7] Alex Huth, Ravindra Duddu, Benjamin Smith, and Olga Sergienko. Simulating the processes controlling ice-shelf rift paths using damage mechanics. *Journal of Glaciology*, pages 1–14, September 2023.

[8] Arne Keller and Kolumban Hutter. Conceptual thoughts on continuum damage mechanics for shallow ice shelves. *Journal of Glaciology*, 60(222):685–693, 2014.

[9] O Mahrenholtz and Z Wu. Determination of creep damage parameters for polycrystalline ice. In *Third International Conference on Ice Technology, Advances in Ice Technology*, Massachusetts Institute of Technology, Cambridge, MA, 1992.

[10] F.M. Nick, C.J. Van Der Veen, A. Vieli, and D.I. Benn. A physically based calving model applied to marine outlet glaciers and implications for the glacier dynamics. *Journal of Glaciology*, 56(199):781–794, 2010.

[11] John Nye. The distribution of stress and velocity in glaciers and ice-sheets. *Proceedings of the Royal Society of London. Series A. Mathematical and Physical Sciences*, 239(1216):113–133, February 1957.

[12] A. Pralong and M. Funk. Dynamic damage model of crevasse opening and application to glacier calving. *Journal of Geophysical Research*, 110(B1):B01309, 2005.

[13] Sainan Sun, Stephen L. Cornford, John C. Moore, Rupert Gladstone, and Liyun Zhao. Ice shelf fracture parameterization in an ice sheet model. *The Cryosphere*, 11(6):2543–2554, November 2017.

---

## Author Response (AR2)

The comments from the reviewer are in black font and our responses are in blue font.

I would like to thank the authors for their thorough response to both my main concerns and smaller comments. I believe that the revisions made to the manuscript will help the readers better understand the proposed damage model and its implications (e.g., plotting Figure 2 for a 1000-year simulation).

We absolutely agree that the recommendations by both reviewers and the editor significantly strengthened this paper, and we appreciate all the time and effort on the part of the reviewers.

As I previously commented, I believe the SSA limits the realism of the current model since it does not account for velocity and stress distribution over the ice column, and consequently, cannot capture vertical advection of damage. These limitations are acknowledged in the discussion, and I understand that this is beyond the initial scope of this work, which aims to introduce ?a simple damage model to simple ice flow models.? Using a higher-order model and comparing the results with the SSA would be an interesting follow-up study, which the authors seem to have considered, and which I look forward to read!

We also agree that the SSA limitations do hinder a full representation of some potentially important damage physics (such as, as the reviewer mentions, vertical damage advection), and we are eager in follow-up work to explore the implications of using a model that captures more of the three-dimensional behavior.

Upon- re-reading the manuscript, I came across the following sentence, and I would be interested to hear the authors? thoughts on it:

?The loss of load-bearing capacity in a continuous region across the ice shelf means that downstream ice transmits no buttressing stress upstream, thus representing the dynamic effect of calving on the remaining ice, even in the absence of explicitly simulated iceberg detachment. Some damage also accumulates in the margins near the grounding line and in the center of the ice shelf.?

I am not suggesting that this be discussed in the manuscript, but I wonder: for a model like yours, there do you think there is little need to explicitly simulate a proper calving front (like in CalvingMIP for example)? Since most of highly damaged areas, which should calve, have almost no impact on upstream flow?

This is an interesting point, and I think to some degree is correct; in our model, the "calving front" is delineated by the boundary between ice that has lost nearly all load-bearing ability (with full-thickness damage near 1) and ice that still retains some load-bearing ability and thus ability to transmit buttressing force upstream. The key would be to ensure that $D_{\max}$ is as close to 1 as possible to ensure very little force is transmitted upstream by regions that are supposed to be maximally fractured. The other question would be how to handle the ice-ocean interface; since, in our model, the maximally-damaged ice still exists, that ice is still interacting with the ocean and the true calving front (where the ice still has some load-bearing ability) is not engaging with the ocean at its terminus in the same way it should if the maximally-damaged ice was removed. I imagine particularly in coupled ice-ocean simulations, this would have significant implications to the dynamics of the ice sheet. Therefore, I think there are situations where actually simulating a moving calving front will be important, but in some studies (if you were interested in focusing on the effect of fracture on buttressing, for example), this may be a reasonable way of handling calving.

Technical suggestions:

    Line 383: typo ?cressvassing? → ?crevassing?

    Line 443: ?to to?

    Line 587: The ice sheet model icepack uses the Shallow-Shelf Approximation (SSA), which calculates depth-averaged flow fields [?]

Done.